# Learning General World Models in a Handful of Reward-Free Deployments

**Yingchen Xu**[*]
UCL, FAIR

**Jack Parker-Holder**[*]
University of Oxford

**Aldo Pacchiano**[*]
Microsoft Research

**Philip J. Ball**[*]
University of Oxford

**Oleh Rybkin**
UPenn

**Stephen J. Roberts**
University of Oxford

**Tim Rocktäschel**
UCL

**Edward Grefenstette**
UCL, Cohere

## Abstract

Building generally capable agents is a grand challenge for deep reinforcement learning (RL). To approach this challenge practically, we outline two key desiderata: 1) to facilitate generalization, exploration should be task agnostic; 2) to facilitate scalability, exploration policies should collect large quantities of data without costly centralized retraining. Combining these two properties, we introduce the *reward-free deployment efficiency* setting, a new paradigm for RL research. We then present CASCADE, a novel approach for self-supervised exploration in this new setting. CASCADE seeks to learn a world model by collecting data with a population of agents, using an information theoretic objective inspired by Bayesian Active Learning. CASCADE achieves this by specifically maximizing the *diversity of trajectories* sampled by the population through a novel *cascading* objective. We provide theoretical intuition for CASCADE which we show in a tabular setting improves upon naïve approaches that do not account for population diversity. We then demonstrate that CASCADE collects diverse task-agnostic datasets and learns agents that generalize zero-shot to novel, unseen downstream tasks on Atari, MiniGrid, Crafter and the DM Control Suite. Code and videos are available at https://ycxuyingchen.github.io/cascade/

## 1 Introduction

Reinforcement learning (RL, [105]) has achieved a number of impressive feats over the past decade, with successes in games [69, 13, 100], robotics [48, 77], and the emergence of real world applications [11, 25]. Indeed, now that RL has successfully mastered a host of individual tasks, the community has begun to focus on the grand challenge of building *generally capable* agents [90, 109, 68, 5].

In this work, we take steps towards building *generalist agents at scale*, where we outline two key desiderata. First, for agents to become generalists that can adapt to novel tasks, we eschew the notion of restricting agent learning to task-specific reward functions and focus on the *reward-free* problem setting instead [83, 28], whereby agents must discover novel skills and behaviors without supervision.[2] Consider the problem of learning to control robotic arms, where we may already have some expert offline data to learn from. In many cases this data will cover only a subset of the entire range of possible behaviors. Therefore, to learn additional general skills, it is imperative to collect additional novel and diverse data, and to do so without a pre-specified reward function.

Second, to ensure scalability, we should have access to a large fleet of robots that we can *deploy* to gather this data for a large number of timesteps [60], without costly and lengthy centralized

---

[*]Equal contribution. Correspondence to ycxu@meta.com.

[2]Indeed, designing reward functions to learn behaviors can have unintended consequences [74, 6].

36th Conference on Neural Information Processing Systems (NeurIPS 2022).

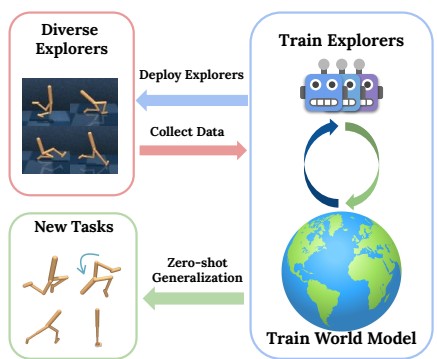
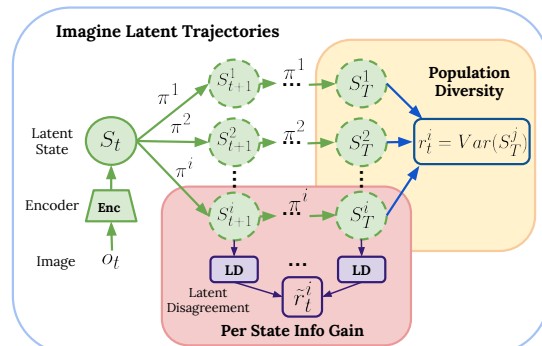

Figure 1: **Overview.** Left: CASCADE trains a population of diverse explorers and uses them to collect large batches of reward-free trajectories for learning a general world model that facilitates zero-generalization to novel tasks. Right: To train $B$ exploration agents in parallel, at each training step $t$, CASCADE first infers a latent state $s_t$ from image observation $o_t$. It then rolls out latent trajectories $\tau^1, \ldots, \tau^B$ in imagination using the current exploration policies $\pi^1, \ldots, \pi^B$. The training objective for each policy $\pi^i$ is to optimize 1) the population diversity estimated by the disagreement of the final states of imagined trajectories sampled from policies $\pi^1, \ldots, \pi^i$; 2) the expected per state information gain over all future timesteps $t+1, \ldots, T$, computed as the disagreement of an ensemble of dynamics models.

retraining during this crucial phase [67]. This has recently been referred to as *deployment efficiency*, falling between the typical online/offline RL dichotomy. Limiting deployments not only reduces the overhead in retraining exploration policies, but also limits the potential costs and risks present when deploying new policies [108, 53], an important consideration in many real world settings, such as robotics [48], education [65] and healthcare [29].

Combining these two desiderata, we introduce the *reward-free deployment efficiency* setting, a new paradigm for deep RL research. To tackle this new problem we train a population of exploration policies to collect large quantities of useful data via *world models* [20, 34, 39]. World models allow agents to plan and/or train policies without interacting with the true environment. They have already been shown to be highly effective for deployment efficiency [67], offline RL [118, 49, 8, 88] and self-supervised exploration [99, 97]. Furthermore, world models offer the potential for increasing agent generalization capabilities [7, 37, 68, 16, 8, 21, 10, 58], one of the frontiers of RL research [80, 51]. However, since existing self-supervised methods for learning world models are designed to collect only a few transitions with a single exploration policy, they likely produce a *homogenous dataset* when deployed at scale, which does not optimally improve the model.

Instead, drawing analogies from Bayesian Active Learning [42, 52], we introduce a new information theoretic objective that maximizes the information gain from an *entire dataset* collected by a *population* of exploration agents (see Figure 2). We call our method *Coordinated Active Sample Collection via Diverse Explorers* or CASCADE (Figure 1). We provide theoretical justification for CASCADE, which emphasizes the importance of collecting data with diverse agents. In addition, we provide a rigorous empirical evaluation across four challenging domains that shows CASCADE can discover a rich dataset from a handful of deployments. We see that CASCADE produces general exploration strategies that are equally adept at both "deep" exploration problems and diverse behavior discovery. This makes it possible to train agents capable of zero-shot transfer when rewards are provided at test time in a variety of different settings.

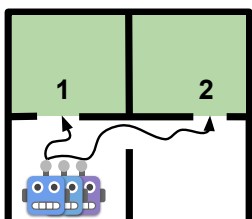

Figure 2: Motivation for CASCADE: Green areas represent high expected information gain. If we train a population of agents independently, at deployment time they will all follow the trajectory to #1, producing a homogenous dataset. However, if we consider the diversity of the data then we will produce agents that reach both #1 and #2.

To summarize, our contributions are as follows: 1) We introduce a novel problem setting, *Reward-Free Deployment Efficiency*, designed to train generalist agents in a scalable fashion; 2) We propose CASCADE, a theoretically motivated model-based RL agent designed to gather diverse, highly informative data, inspired by Bayesian Active Learning; 3) We provide analysis that shows CASCADE theoretically improves sample efficiency over other naïve methods that do not ensure sample diversity, and demonstrate that CASCADE is capable of *improved zero-shot transfer* in four distinct settings, ranging from procedurally generated worlds to continuous control from pixels.

## 2 Problem Statement

Reinforcement learning (RL) considers training an agent to solve a Markov Decision Process (MDP), represented as a tuple $\mathcal{M} = \{\mathcal{S}, \mathcal{A}, P, R, \rho, \gamma\}$, where $s \in \mathcal{S}$ and $a \in \mathcal{A}$ are the set of states and actions respectively, $P(s'|s,a)$ is a probability distribution over next states given a previous state and action, $R(s, a, s') \rightarrow r$ is a reward function mapping a transition to a scalar reward, $\rho$ is an initial state distribution and $\gamma$ is a discount factor. A policy $\pi$ acting in the environment produces a trajectory $\tau = \{s_1, a_1, \ldots, s_H, a_H\}$ for an episode with horizon $H$. Since actions in the trajectory are sampled from a policy, we can then define the RL problem as finding a policy $\pi$ that maximizes expected returns in the environment, i.e. $\pi^\star = \arg\max_\pi \mathbb{E}_{\tau \sim \pi}[R(\tau)]$.

We seek to learn policies that can transfer to *any* MDP within a family of MDPs. This can be formalized as a *Contextual* MDP [51], where observations, dynamics and rewards can vary given a context. In this paper we consider settings where only the reward varies, thus, if the test-time context is unknown at training time we must collect data that *sufficiently covers the space of possible reward functions*. Finally, to facilitate scalability, we operate in the deployment efficient paradigm [67], whereby policy learning and exploration are completely separate, and during a given *deployment*, we gather a large quantity of data without further policy retraining (c.f. online approaches like DER [112], which take multiple gradient steps *per* exploration timestep in the real environment). Taken together, we consider the *reward-free deployment efficiency* problem. This differs from previous work as follows: 1) unlike previous deployment efficiency work, our exploration is task agnostic; 2) unlike previous reward-free RL work, we cannot update our exploration policy $\pi_{\mathrm{EXP}}$ during deployment. Thus, the focus of our work is on how to train $\pi_{\mathrm{EXP}}$ offline such that it gathers heterogeneous and informative data which facilitate zero-shot transfer to unknown tasks.

In this paper we make use of model-based RL (MBRL), where the goal is to learn a model of the environment (or *world model* [96]) and then use it to subsequently train policies to solve downstream tasks. To do this, the world model needs to approximate both $P$ and $R$. Typically, the model will be a neural network, parameterized by $\psi$, hence we denote the approximate dynamics and reward functions as $P_\psi$ and $R_\psi$, which produces a new "imaginary" MDP, $\mathcal{M}_\psi = (\mathcal{S}, \mathcal{A}, P_\psi, R_\psi, \rho)$. We focus on Dyna-style MBRL [104], whereby we train a policy ($\pi_\theta$ parameterized by $\theta$) with model-free RL solely using "imagined" transitions inside $\mathcal{M}_\psi$. Furthermore, we can train the policy on a single GPU with parallelized rollouts since the simulator is a neural network [54]. The general form of all methods in this paper is shown in Algorithm 1, with the key difference being step 5: We aim to update $\pi_{\mathrm{EXP}}$ in the new imaginary MDP $\mathcal{M}_\psi$ such that it continues to collect a large, diverse quantity of reward-free data. Note that $\pi_{\mathrm{EXP}}$ need not be a single policy, but could also refer to a collection of policies that we can deploy (either in parallel or in series), such that $\pi \in \pi_{\mathrm{EXP}}$.

---

**Algorithm 1** Reward-Free Deployment Efficiency via World Models

---

1: **Input:** Initial exploration policy $\pi_{\mathrm{EXP}}$
2: **for** each deployment **do**
3:      Deploy $\pi_{\mathrm{EXP}}$ to collect a *large quantity* of *reward-free* data.
4:      Train world model on all existing data.
5:      Update $\pi_{\mathrm{EXP}}$ in new imaginary MDP $\mathcal{M}_\psi$.
6: **end for**

---

We focus on learning world models from high dimensional sensory inputs such as pixels [34, 76, 47], where at each timestep we are given access to an observation $o_t$ rather than a state $s_t$. A series of recent works have shown tremendous success by mapping the observation to a compact latent state $z_t$ [39, 38, 40]. In this paper we will make use of the model from DreamerV2 [40], which has been shown to produce highly effective policies in a variety of high dimensional environments. The primary component of DreamerV2 is a Recurrent State Space Model (RSSM) that uses a learned latent state to predict the image reconstruction, reward $r_t$ and discount factor $\gamma_t$. Aside from the reward head, all components of the model are trained jointly, in similar fashion to variational encoders (VAEs, [50, 92]). For zero-shot evaluation, we follow [97] and only train the reward head at test time when provided with labels for our pre-collected data, which is then used to train a behavior policy offine. Thus, it is critical that our dataset is sufficiently diverse to enable learning novel, unseen behaviors.

## 3 Coordinated Active Sample Collection

The aim of this work is to train a population of $B$ exploration policies $\{\pi_{\text{EXP}}^{(i)}\}_{i=1}^{B}$ such that they collectively acquire data which maximally improves the accuracy of a *world model*. To achieve this, we take inspiration from the information theoretic approach in Plan2Explore [97], but crucially focus on maximizing information gain over *entire trajectories* rather than per state-action, and hence drop the conditional dependence on state and action (see App. C.1 for why this distinction is important):

$$\pi_{\text{EXP}} = \arg\max_{\pi} \mathcal{I}\left(d_{\mathcal{M}_\psi}^\pi; \mathcal{M}_\psi\right) = \mathcal{H}(d_{\mathcal{M}_\psi}^\pi) - \mathcal{H}(d_{\mathcal{M}_\psi}^\pi | \mathcal{M}_\psi) \tag{1}$$

where $d_{\mathcal{M}_\psi}^\pi$ is the distribution of states visited by the policy $\pi$ in the imaginary MDP $\mathcal{M}_\psi$. This objective produces $\pi_{\text{EXP}}$, a policy whose visitation distribution has a high entropy when computed over model samples, but has low entropy for each individual MDP model (i.e., high epistemic/reducable uncertainty). We think of each model $\mathcal{M}_\psi$ as sampled from a posterior distribution over models given the data. A good exploration policy has low entropy on individual models but large entropy across models, i.e. it is intent in visiting regions of the space where there is large uncertainty about the model transitions. To make this objective more general, we represent the trajectory data collected by a policy with a "summary" embedding space [79, 72]. Let $\Phi : \Gamma \to \Omega$ be a summary function mapping trajectories into this embedding space. $\mathbb{P}_\pi^\Phi[\mathcal{M}_\psi]$ denotes the embedding distribution generated by policy $\pi$ in imaginary MDP $\mathcal{M}_\psi$. We can now write the objective from Eq. 1 as follows:

$$\pi_{\text{EXP}} = \arg\max_{\pi} \mathcal{I}\left(\mathbb{P}_\pi^\Phi[\mathcal{M}_\psi]; \mathcal{M}_\psi\right) = \mathcal{H}(\mathbb{P}_\pi^\Phi[\mathcal{M}_\psi]) - \mathcal{H}(\mathbb{P}_\pi^\Phi[\mathcal{M}_\psi] | \mathcal{M}_\psi) \tag{2}$$

This more general framework allows us to consider multiple representations for trajectories, in a similar fashion to behavioral characterizations in Quality Diversity algorithms [85]. For the rest of this discussion, we will use the final state embedding as our summary representation, whereby $\Phi(\tau) = h_H$, since in the case of the RSSM, the final latent is a compact representation of the entire trajectory collected by the policy, analogous to the final hidden state in an RNN [94, 103].

### 3.1 A Cascading Objective with Diverse Explorers

We now consider a population-based version of Equation. 2, using $B$ agents:

$$\{\pi_{\text{EXP}}^{(i)}\}_{i=1}^{B} = \arg\max_{\boldsymbol{\pi}^B \in \Pi^B} \mathcal{I}\left(\prod_{i=1}^{B} \mathbb{P}_{\pi^{(i)}}^\Phi[\mathcal{M}_\psi]; \mathcal{M}_\psi\right) = \mathcal{H}\left(\prod_{i=1}^{B} \mathbb{P}_{\pi^{(i)}}^\Phi[\mathcal{M}_\psi]\right) - \mathcal{H}\left(\prod_{i=1}^{B} \mathbb{P}_{\pi^{(i)}}^\Phi[\mathcal{M}_\psi] \Big| \mathcal{M}_\psi\right) \tag{3}$$

where $\boldsymbol{\pi}^B = \pi^{(1)}, \cdots, \pi^{(B)}$ and $\prod_{i=1}^{B} \mathbb{P}_{\pi^{(i)}}^\Phi[\mathcal{M}_\psi]$ is the product measure of the policies' embedding distributions in $\mathcal{M}_\psi$. By definition, the conditional entropy factorizes as:

$$\mathcal{H}\left(\prod_{i=1}^{B} \mathbb{P}_{\pi^{(i)}}^\Phi[\mathcal{M}_\psi] \Big| \mathcal{M}_\psi\right) = \sum_{i=1}^{B} \mathcal{H}\left(\mathbb{P}_{\pi^{(i)}}^\Phi[\mathcal{M}_\psi] \Big| \mathcal{M}_\psi\right). \tag{4}$$

It is now possible to show that maximum information gain is achieved with a *diverse* set of agents:

**Lemma 1.** *When all models $\mathcal{M}_\psi$ in the support of the model posterior are deterministic and tabular, and the space of policies $\Pi$ consists only of deterministic policies, there always exists a solution $\{\pi_{\text{EXP}}^{(i)}\}_{i=1}^{B}$ satisfying $\pi_{\text{EXP}}^{(i)} \neq \pi_{\text{EXP}}^{(j)} \forall i \neq j$. Moreover, there exists a family of tabular MDP models, such that the maximum cannot be achieved by setting $\pi_{\text{EXP}}(i) = \pi$ for a fixed $\pi$.*

The proof of Lemma 1 is in Appendix C.1.1. Since the mutual information objective 3 is submodular, a greedy algorithm yields a $(1 - \frac{1}{e})$ approximation of the optimum (where $e$ is Euler's number) [73]. Leveraging this insight, let us assume that we already have a set of policies $\pi^{(1)}, \cdots, \pi^{(i-1)}$; we then select the next policy $\pi^{(i)}$ based on the following greedy objective:

$$\pi^{(i)} = \arg\max_{\tilde{\pi}^{(i)} \in \Pi} \mathcal{I}\left(\prod_{j=1}^{i} \mathbb{P}_{\tilde{\pi}^{(j)}}^\Phi[\mathcal{M}_\psi]; \mathcal{M}_\psi \Big| \tilde{\pi}^{(j)} = \pi^{(j)} \ \forall j \leq i-1\right)$$

$$= \mathcal{H}\left(\prod_{j=1}^{i} \mathbb{P}_{\tilde{\pi}^{(j)}}^\Phi[\mathcal{M}_\psi] \Big| \tilde{\pi}^{(j)} = \pi^{(j)} \ \forall j \leq i-1\right) - \mathcal{H}\left(\prod_{j=1}^{i} \mathbb{P}_{\pi^{(j)}}^\Phi[\mathcal{M}_\psi] \Big| \mathcal{M}_\psi, \tilde{\pi}^{(j)} = \pi^{(j)} \ \forall j \leq i-1\right)$$

Which can be factorized in similar fashion to Equation 4 (See Appendix C).

## 3.2 A Tractable Objective for Deep RL

Inspired by [97], we make a couple of approximations to derive a tractable objective for $\{\pi_{\text{EXP}}\}_{i=1}^{B}$ in the deep RL setting. First, we assume that the final state embedding distributions are Gaussian with means that depend on the policies and sampled worlds, and variances that depend on the worlds, i.e. $\mathbb{P}_{\pi}^{\Phi}[\mathcal{M}_{\psi} = w] = \mathcal{N}(\mu(w, \pi), \Sigma(w))$ . In this case, $\mathcal{H}(\mathbb{P}_{\pi}^{\Phi}[\mathcal{M}_{\psi}|\mathcal{M}_{\psi} = w) = \rho(w)$, and Eq. 5 reduces to solving $\pi^{(i)} = \arg\max_{\tilde{\pi}^{(i)} \in \Pi} \mathcal{H}\left(\prod_{j=1}^{i} \mathbb{P}_{\tilde{\pi}^{(j)}}^{\Phi}[\mathcal{M}_{\psi}]\Big|\tilde{\pi}^{(j)} = \pi^{(j)} \ \forall j \leq i - 1\right)$ for a policy that maximizes the resulting joint entropy of the embedding distribution when added to the policy population. This produces the following surrogate objective, maximizing a quadratic *cascading* disagreement:

$$\text{PopDiv}^{\Phi}(\pi|\pi^{(1)}, \cdots, \pi^{(i-1)}) = \mathbb{E}_{\tau \sim \mathbb{P}^{\pi}[\mathcal{M}_{\psi}]}\left[\frac{1}{|\mathcal{D}^{(i-1)}| - 1}\sum_{\tilde{\tau} \in \mathcal{D}^{(i-1)}}\|\Phi(\tau) - \Phi(\tilde{\tau})\|^2\right]$$

where $\mathcal{D}^{(i-1)}$ is a dataset of imagined trajectories sampled from policies $\pi^{(1)}, \cdots, \pi^{(i-1)}$ in the model, and PopDiv is short for Population Diversity. Finally, following [97, 9], we also add a per state information gain component to each policy's reward to encourage a richer landscape for data acquisition: $\text{InfoGain}(\pi) = \mathbb{E}_{\tau \sim \mathbb{P}^{\pi}[\mathcal{M}_{\psi}]}\left[\sum_{(s,a) \in \tau} \sigma(s, a)\right]$ where $\sigma(\cdot, \cdot)$ is the variance across the ensemble latent state predictions (for details see App. B).

Taken together, these objectives form our approach, which we call *Coordinated Active Sample Collection via Diverse Explorers* or CASCADE. CASCADE trains agents to optimize: 1) a diversity term (PopDiv) that takes into account the behaviors of the other agents in the population; 2) an information gain term (InfoGain) that encourages an individual agent to sample states that maximally improve the model:

$$\pi^{(i)} = \arg\max_{\pi \in \Pi}\left[\lambda \text{PopDiv}^{\Phi}(\pi|\{\pi_{\text{EXP}}^{(j)}\}_{j=1}^{i-1}) + (1 - \lambda)\text{InfoGain}(\pi)\right] \tag{6}$$

where $\lambda$ is a weighting hyperparameter that trades off whether we favor individual model information gain or population diversity. Finally, we train the $B$ agents in *parallel* using (policy) gradient descent over $\theta$, which makes it possible to achieve the same wall clock time as training a single agent [23].

## 3.3 Theoretical Motivation

We now seek to provide a tabular analogue to CASCADE which provides a theoretical grounding for our approach. In App. D we outline the pseudo-code of CASCADE-TS, a greedy Thompson Sampling algorithm [4] that produces the $i$-th exploration policy in a tabular enviornment using "imaginary" data gathered by running policies $\pi^{(1)}, \cdots, \pi^{(i-1)}$ in the *model*. We can then show the following:

**Lemma 2.** *For the class of Binary Tree MDPs, the CASCADE-TS algorithm satisfies,*

$$T(\epsilon, \text{Sequential}) \leq T(\epsilon, \text{CASCADE-TS}) \leq T(\epsilon, \text{SinglePolicyBatch})$$

*where $T(\epsilon, \cdot)$ are the expected number of rounds of deploying a population of $B$ policies necessary to learn the true model up to $\epsilon$ accuracy;* SinglePolicyBatch *plays a fixed policy $B$ times in each round;* Sequential *does not have a population, and instead interleaves updates and executions of a single policy $B$ times within each round.*

The proof is in App. D. Indeed, we see that CASCADE-TS achieves provable efficiency gains over a naïve sampling approach that does not ensure diversity in its deployed agents. This provable gain is achieved by discouraging the $i$-th policy away from imaginary state-action pairs sampled by the previous $i - 1$ policies by using imaginary counts.

Now returning to CASCADE, we can see the importance of leveraging the imaginary data gathered in the model by the previous $i - 1$ policies when training the $i$-th exploration policy. Concretely, encouraging policy $\pi^{(i)}$ to induce high disagreement with the embeddings produced by $\{\pi^{(1)}, \cdots, \pi^{(i-1)}\}$ (i.e. the PopDiv term) is analogous to the imaginary count bonus term of CASCADE-TS in Line 10 of Alg. 2, which avoids redundant data collection during deployment.

# 4 Experiments

In our experiments we test whether CASCADE facilitates the learning of generalist agents by evaluating their zero-shot transfer to unseen tasks, given a limited number of reward-free deployments. In all experiments, agents cannot train online in the environment, and instead execute a fixed exploration policy to gather new data during each deployment. We test against three baselines. The first is random exploration, which is used in the majority of RL publications. Our next baseline is Plan2Explore [97] (P2E), which trains a single exploration policy that optimizes for expected information gain inside a world model. We then include a population-based version of P2E, which we call *Population Plan2Explore* (PP2E). PP2E trains a set of randomly initialized agents that independently seek to maximize expected information gain. All methods make use of a DreamerV2 world model [40] and use the same hyperparameters for model and agent training (more details in App. B).

We test all methods in two separate settings: Firstly *Exploring Worlds* (Sec 4.1), where we seek to collect data in challenging exploration environments. We test how well the explorers cover the state space and discover sparse rewards (also known as "deep exploration" [78, 75]). Secondly, we consider *Exploring Behaviors* (Sec 4.2) where we consider collecting data in continuous control environments. In both settings we test zero-shot transfer as follows: 1) we provide reward labels to the model; 2) we train a separate reward head; 3) we train a task specific behavior policy and test it in the environment. This tests whether our model is general enough to facilitate learning downstream policies for previously unknown tasks [97, 55, 116].

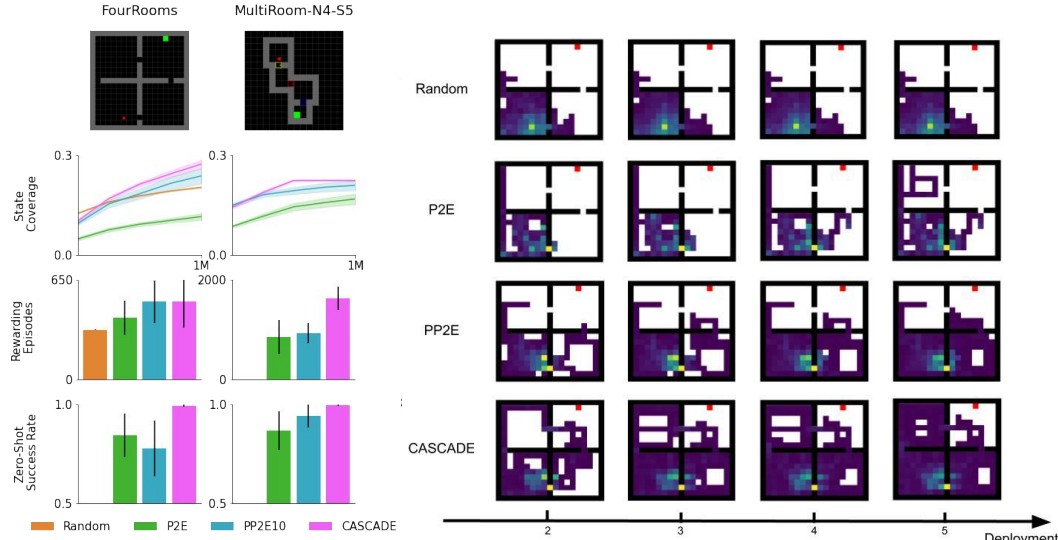

Figure 3: **MiniGrid Results**. Left: Performance statistics for both `FourRooms` and `MultiRoom`. For reward-free exploration we show both "state coverage", which corresponds to the the percentage of states visited on a fixed set of levels, and "rewarding episodes" which is the count of total solved episodes found after 1M steps. Finally for zero-shot transfer we show the success rate of a task-specific behavior policy trained with labels provided only at test time. All plots show the mean of ten seeds with SEM shaded. (Random achieves 0 zero-shot success rate). Right: we show the state coverage on a fixed level in `FourRooms`, using the exploration policy $\pi_{\text{EXP}}$ for each deployment (Test time goal state indicated by the red dot).

## 4.1 Exploring Worlds

In many cases we may not know a priori which states in an environment are highly rewarding. Thus, if we wish to use our world model to subsequently train agents to solve tasks, it is crucial that we accurately model as much of state space as possible. Here we consider exploring worlds in three different environments: MiniGrid [19], Atari [12] and Crafter [37]. In all three settings, we start with a *random* deployment before beginning reward-free exploration.

We begin with MiniGrid, a set of sparse-reward, partially observable navigation environments commonly used to test state-of-the-art exploration methods [89, 17, 31, 121]. MiniGrid environments are procedurally generated, which provides an additional generalization challenge [46, 93]. We

consider the canonical `FourRooms` and the more challenging `MultiRoom`. In each case we are limited to 5 deployments of 200k transitions, for a total of 1M environment interactions.[3] In order to gauge the effectiveness of the reward-free exploration, we use two metrics: 1) state coverage, where we evaluate the percentage of the states visited by the exploration policies after each deployment on the same set of held out test levels; and 2) rewarding episodes, where we show the number of rewarding episodes collected during the training process. Note that these two objectives are not useful in isolation, as solved levels may only include a narrow set of observations, while state coverage alone could be maximized without performing deep, goal-seeking exploration in the environment. However, in combination they provide a proxy for exploration.

We show the results in Figure 3, where we see the CASCADE exploration policies cover the state space far more effectively than all baselines. In `FourRooms`, both CASCADE and PP2E find the goal an equal number of times, while CASCADE finds almost double the number of rewarding episodes in the `MultiRoom` environment. We also test zero-shot performance, where CASCADE achieves almost 100% success rate. Interestingly, in both settings the Random policy does cover a large proportion of the state space, but finds fewer rewarding episodes and subsequently does not have a sufficient quantity to train a policy to solve the task (and accordingly achieves 0% success rate).

Next we consider four Atari games: `Montezuma's Revenge`, `Frostbite`, `Hero` and `Freeway`, often used as barometers for exploration capabilities [27]. We use 15 reward-free deployments, each consisting of 200k steps, and use the final model to train a behavior policy with task reward. We test these agents in a zero-shot fashion, with results shown in the first three columns of Figure 4. We see that CASCADE gets statistically significantly stronger zero-shot returns compared to the baselines. Furthermore, the baseline methods display no clear trend, with random exploration sometimes offering the strongest performance. On the other hand, we see that CASCADE performs consistently well, providing further evidence of it being a strong general exploration strategy. See App. A.2 for more detailed analysis.

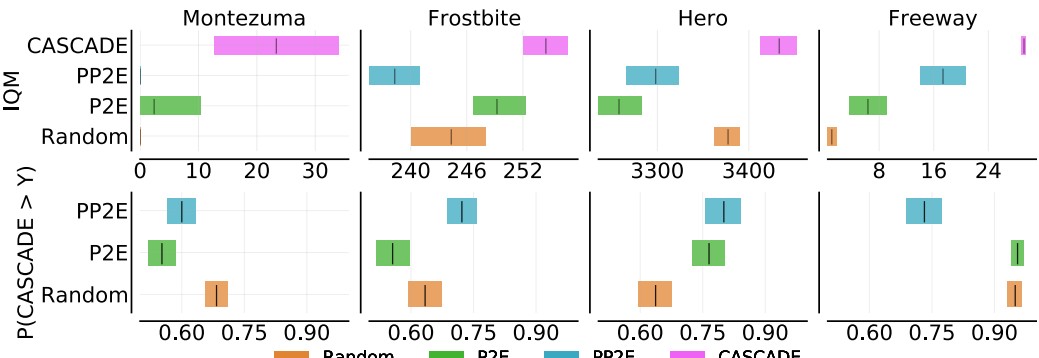

Figure 4: **Atari Zero-Shot Transfer**: Plots show RLiable metric performance aggregated across all seeds. Note that the 'Probability of Improvement' lower CIs all exceed 0.5 for Atari, indicating CASCADE provides a statistically significant improvement (under the Mann-Whitney U test) over all baseline methods [3]. We show five seeds of zero-shot test performance.

Finally, we consider the `Crafter` environment [37], a procedurally generated grid world based on the game of Minecraft. This more complex setting requires the agent to master a variety of compositional skills to fully explore all possible behaviors. Online P2E represents the state of the art reward-free agent, achieving a "Crafter Score" of 2.1. Making the problem more challenging, we deploy only 20 distinct exploration policies, each collecting 50k timesteps, which leads to weaker performance for P2E vs. the reported result in [40]. However, as we see in Figure 5, adding a population of diverse explorers recovers the majority of this performance. We find this result encouraging as the behaviors required to achieve a higher Crafter score are non-trivial. Moreover, as shown in Figure 6, CASCADE achieves the highest average success rate and unlocks the most number of unique Crafter skills among all baselines, including Making a Stone Pickaxe. We believe this opens the door to further gains with more active tuning of the behavioral representation used in CASCADE.

---

[3]Zero-shot success rates of P2E, PP2E and CASCADE increase with more training steps, and CASCADE achieves a near 100% success rate at 1M steps in both `FourRooms` and `MultiRoom`.

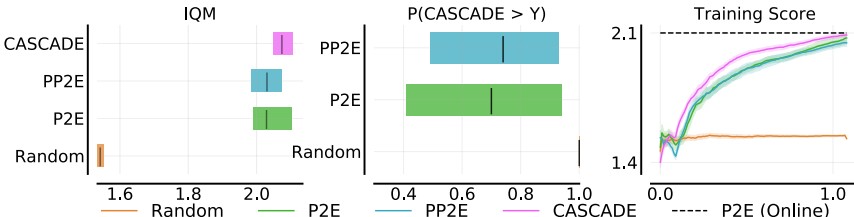

Figure 5: **Crafter Score**: Plots show final RLiable metric performance and training curve over 1M steps aggregated across all seeds. We show ten seeds of Crafter Score, the geomean of skills discovered in a purely reward-free fashion. Again, see that the 'Probability of Improvement' lower CIs all exceed 0.5, indicating CASCADE provides a statistically significant improvement (under the Mann-Whitney U test) over all baseline methods [3].

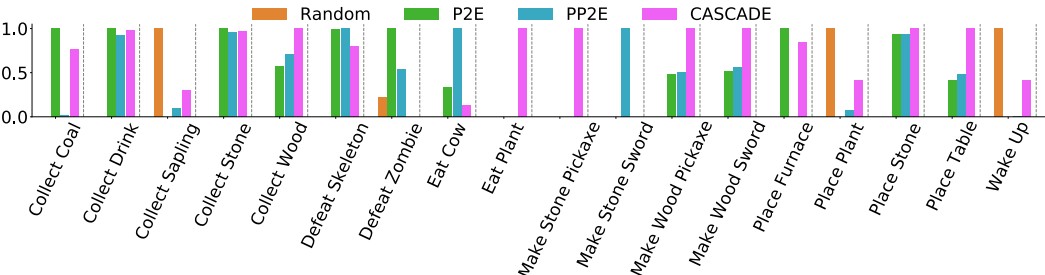

Figure 6: **Crafter Skills**: Plots show normalized success rate per skill. Note that CASCADE not only achieves the highest average success rate but also unlocks the most (16 out of 22) unique skills among all baselines. Skills that are not unlocked by any of the baselines are excluded from the plot.

## 4.2 Exploring Behaviors

Next we wish to test is whether our agents sufficiently explore a range of behaviors in a continuous control setting, using the `walker` environment from the DM Control Suite [111] (DMC). In these experiments, we operate from pixels and assume access to an initial dataset of 200k transitions. Then, we conduct 14 further reward-free deployments, each collecting 200k transitions. We consider the possibility of improving generality from arbitrary offline data, using common datasets from the offline RL literature: `random`, `medium` and `expert`. Using random data represents learning from scratch. In Figure 7 we show the behaviors deployed by CASCADE and PP2E in a single deployment. The CASCADE explorers all display diverse, useful behaviors, while PP2E agents are homogenous.

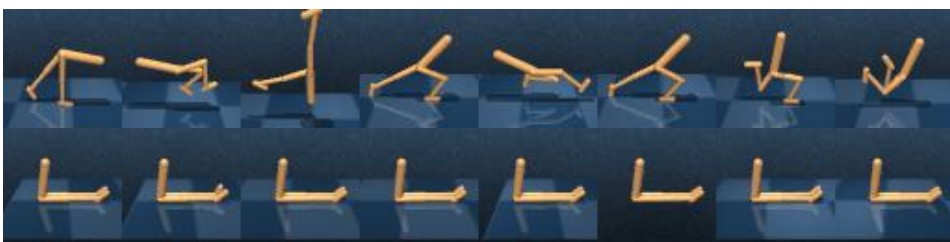

Figure 7: **DMC Qualitative Results**: Each row visualizes behaviors of a population of explorers trained after 15 total deployments by the same world model, deployed at the same time in environments with the same initial state. We can see that CASCADE explorers (top) collect data with a diversity of behaviors while PP2E agents (bottom) only show less interesting homogenous behaviors.

We then test the generality of the world model after each deployment by training a separate agent from scratch for each of the four individual tasks: `stand`, `walk`, `run`, `flip` proposed in URLB [56]. This demonstrates whether the world model is capable of producing an imaginary MDP that facilitates learning specific behaviors. To compare the performance of CASCADE and the baselines, in Figure 8 we show the aggregated statistics from the RLiable library [3] for a given number of deployments. To do this, we combine the results across all datasets (`random`, `medium`, `expert`) and all downstream tasks (`stand`, `walk`, `run`, `flip`) and compute the following: 1) the Inter-Quartile Mean (IQM) of the normalized scores (the robust statistic recommended in [3]); 2) the Probability of Improvement, the likelihood of one method outperforming another on a new, unseen task. As we see, CASCADE shows clear and consistent gains over the next best baseline (PP2E) throughout 15 deployments, showing

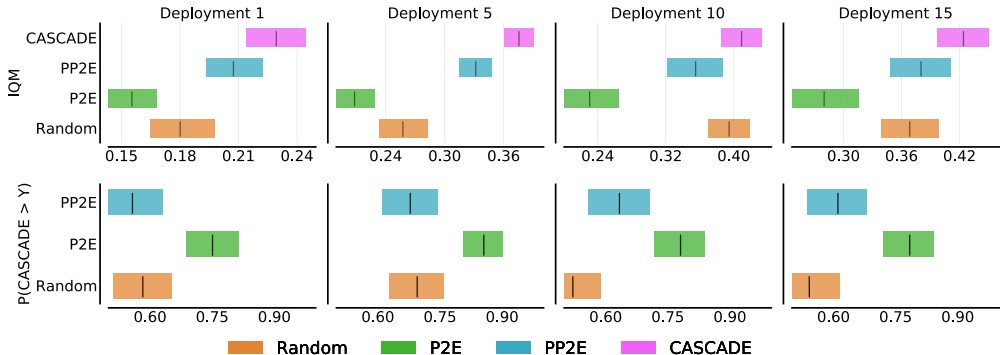

Figure 8: **DMC Zero-Shot Aggregate Results**: Plots show RLiable metric performance aggregated across all three initial datasets (`random`, `medium`, `expert`) for a total of 30 seeds. Note that the 'Probability of Improvement' lower CIs all exceed 0.5, indicating CASCADE provides a statistically significant improvement (under the Mann-Whitney U test) over all baseline methods [3], even after a single deployment.

statistical significance under the Mann-Whitney U test [66]. Note that here the single agent P2E performs poorly, likely due to only collecting data with a single mode of exploratory behavior. See Figure 9 in the Appendix for a full breakdown of the performance.

### 4.3  Discussion and Limitations

We have shown that CASCADE is the most effective method in training generally capable agents across a wide range of environments. Indeed, CASCADE provides the best of both worlds: deep exploration by seeking the frontier of the current knowledge (via InfoGain), while covering the space of possible behaviors (via PopDiv). Each of our baselines lacked in one of these two key properties. Random exploration fared poorly when exploring worlds, which requires deep exploration. However, it can be a highly effective method for exploring behaviors. This is likely why boosting entropy in the action space is so effective for robotics tasks [36], but less so in deep exploration tasks [75]. Furthermore, the information theoretic baselines were effective in Exploring Worlds, but the disappointing performance for P2E in DMC is likely caused by a lack of data diversity, while the PP2E populations were not sufficiently heterogeneous to match the coverage of CASCADE.

In terms of limitations, the main weakness in current methods is the absolute level of performance. Despite outperforming baselines in DMC, the policies learned by CASCADE are far from optimal. This is likely due to the challenge of the setting (reward-free deployment efficiency) and we believe that future work in the community will be crucial to further improve these methods. Finally, CASCADE does require training additional agents that may provide additional computational expense. In this setting we assume the time between deployments is arbitrary, but this may not be the case in all settings and future work could consider ways to make this process more efficient and scalable.

## 5  Related Work

At the core of our paper is the notion of training generalist agents, which have recently been of increased interest in RL [90, 68, 109, 110]. In particular, we focus on the *reward-free* paradigm, since in many cases it may not be possible to know a priori all possible tasks we may wish our agent to solve [45]. With this in mind, there has been a surge of interest in methods that can learn in a self-supervised fashion, to build representations that lead to fast (or even zero-shot) adaptation to future tasks. These methods range from competence-based (i.e. learning "skills") [1, 33, 28, 59, 62, 101] to data-based that aim to maximize data diversity [63, 117].

In this work we take inspiration from *knowledge-based* approaches, which typically seek to learn a world model [9, 83, 84, 30, 107, 15, 98]. In particular we build on Plan2Explore [97] which trains an exploration agent by maximizing an objective resembling information gain inside the world model. Plan2Explore was the first work that showed it is possible to transfer a model learned without rewards in a zero-shot fashion. However, it differed from our work in that Plan2Explore solely considers the online RL setting, deploying a new exploration policy every timestep. Other recent work examined the effectiveness of reward-free RL for collecting datasets for subsequent offline RL [116, 55]. Once

again however, this work trains the exploration agent online. By contrast, we only deploy a handful of exploration agents and collect a larger quantity of data with each one.

Our setting resembles a reward-free version of the *deployment efficiency* paradigm, first introduced in [67]. By contrast, their method focuses on policy optimization in the *supervised* RL paradigm, whereas we focus on exploration and zero-shot generalization in the unsupervised RL setting. Follow up work has remained focused on supervised policy optimization [102], whereas we are focused on training generalist agents in the reward-free paradigm. Recent work has also shown the importance of frequent policy updates in ensuring diverse exploration [95], further supporting the need for a diverse population of explorers when operating in the deployment-efficient setting.

To address the challenges of this novel problem setting, we draw inspiration from a variety of fields. In particular, several works have considered exploration with a population of agents [106, 26, 64], but these have only considered task-specific online RL. Our choice of a diverse population of agents takes inspiration from the field of Quality Diversity (QD, [85, 24]), where it has been shown that diversity can boost exploration in RL [22]. Recently, [61] showed it is possible to train diverse populations inside a world model to acquire new skills. As far as we are aware, we are the first to consider a cascading objective for training diverse populations, which is typically done either synchronously [82] or sequentially [122, 119].

Also related are works that specifically train policies that are able to maximally cover the state space of a given MDP [2, 70]. However, these works focus on the more theoretically amenable (and less general) low-rank MDP setting, and hence do not demonstrate any experiments on larger scale problems. Furthermore, we aim to achieve state-space coverage by encouraging *policy diversity*, which is distinct from the approaches taken in these works. We make use of an information theoretic objective, which has been shown to be highly effective for exploration in RL [71, 43, 41]. Most similar to our work [120] consider the reward free RL setting, and show that maximizing entropy can lead to improved exploration that can facilitate transfer to arbitrary reward functions. By contrast we consider the limited deployment setting and use model-based RL. Finally, our work relates to collective intelligence [35, 115], since each member of the population contributes a small piece to the puzzle that produces a general world model.

# 6 Conclusion

We introduced the *reward-free deployment efficiency* paradigm—an important problem setting for learning generally capable agents in a scalable fashion. To address the challenges of this setting, we proposed CASCADE, a theoretically motivated approach that leverages a diverse population of self-supervised exploration agents. We showed in a variety of experiments that CASCADE produces *general exploration strategies*, adept at both deep exploration *and* gathering sufficiently hetereogeneous datasets that facilitate the learning of diverse downstream behaviors. We believe this work provides a new approach for collecting large, diverse datasets in a self-supervised fashion, inducing a potentially open-ended process for training general world models at scale.

For future work, we could consider alternative means for computing diversity, for example using optimization landscapes [81] or explicit behavioral diversity [28, 82]. These could also be combined with representations that directly incorporate the behaviors of the policy [14] to further improve the design of the "summary" embedding space. Furthermore, we would be interested in considering transformer models [113] for trajectory modelling [18, 44] that ought to scale more gracefully with larger batches of data [87]. Recent work has shown that this class of models can compose unseen complex multi-task behaviors at test time from fixed offline datasets [32], as well as leverage significantly large and diverse offline datasets to improve behavioral modeling [91], highly relevant to both aforementioned desiderata for the reward-free deployment efficiency paradigm.

## Acknowledgments

The authors would like to thank Cong Lu for help with Offline DreamerV2 code, and Minqi Jiang and Robert Kirk for useful discussions. This work was funded by Meta AI.

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
