# Appendix

## A   Additional Experimental Results

### A.1   DMC

Here we show additional experimental results, beginning with a full breakdown of the DMC results for each dataset. In Figure 6 we see the zero-shot transfer performance for models trained with `random`, `medium` and `expert` initial datasets, and 15 subsequent deployments. Interestingly, we see that the `medium` dataset proves to be the most effective for achieving high performance, this may be due to being more diverse than the `expert` dataset, which is a narrow distribution of high performing episodes.

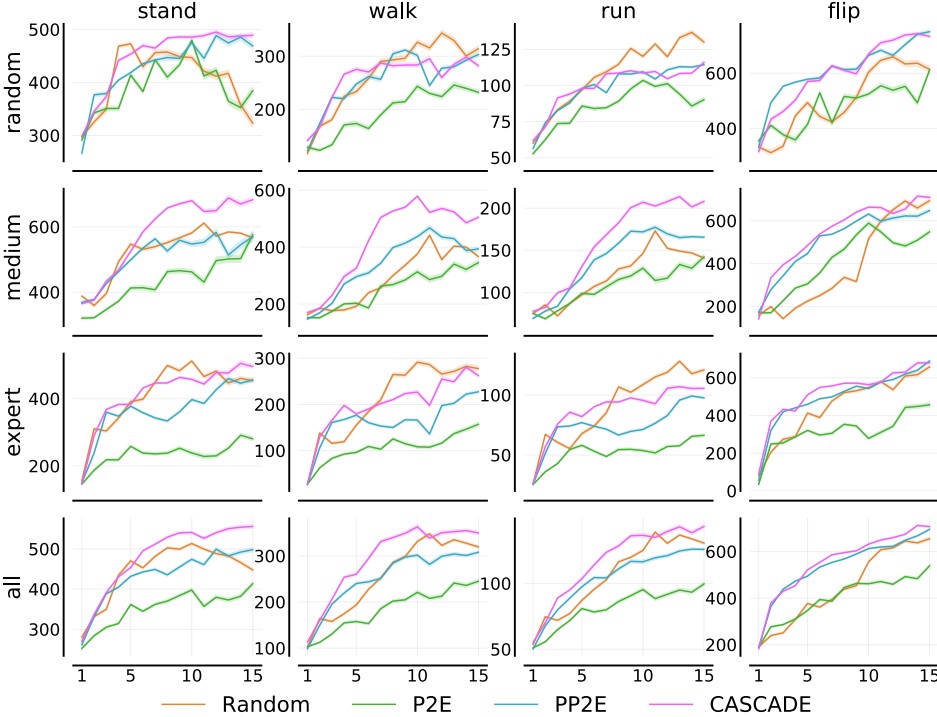

Figure 6: **DMC Zero-Shot Results**: Plots show the performance by environment and dataset, with the last row being the task performance average over all three initial datasets. Plots show the mean and SEM over 10 seeds.

## A.2 Atari Tasks

Next we show detailed results in three Atari games: Montezuma's Revenge, Frostbite and Hero in Fig. 7. We see that generally CASCADE discovers higher rewarding episodes (almost double the next best baseline in Montezuma's Revenge), and also discovers more rewarding episodes on average. This translates to stronger zero-shot performance than all baselines in all three environments. On the other hand, the other baselines achieve worse zero-shot performance because they either (1) find an occasional high reward episode (Max Episode Return), but not in a sufficient quantity (Rewarding Episodes), or (2) collect abundant low rewarding episodes that are less helpful for learning good behavior polices. For instance, the random baseline is able to find Rewarding Episodes more frequently in Frostbite, but these are relatively low quality, hence the lower curve when assessing its Max Episode Return.

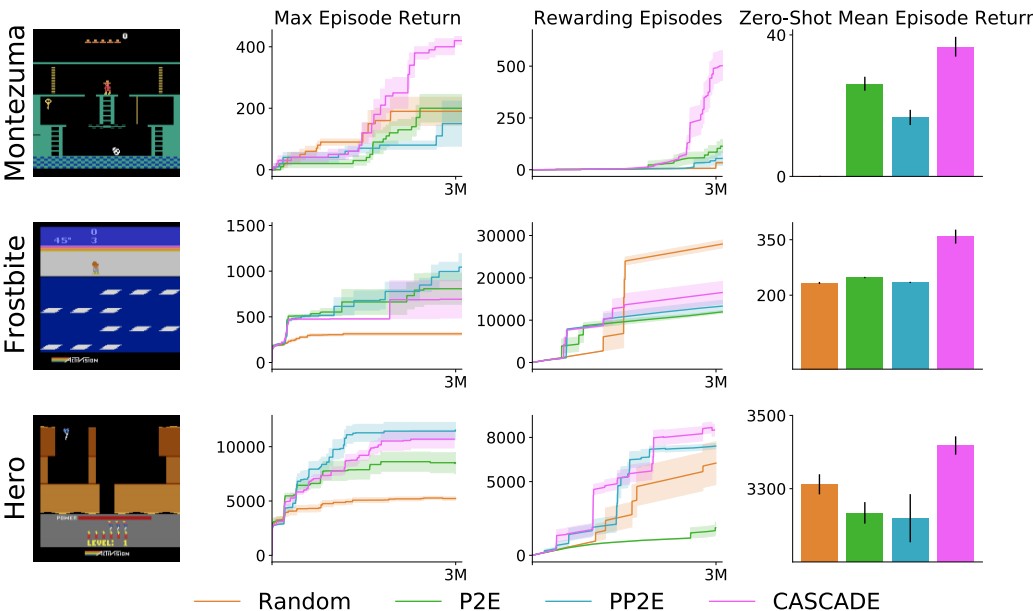

Figure 7: From left to right on each row we show a frame from the game, a plot of the cumulative maximum episode reward, the total number of rewarding episodes discovered and zero-shot average return, from 3M training steps (15 deployments). Plots show mean and SEM over 10 seeds. Note that the max episode returns of all methods stabilize after 15 deployments (3M training steps in total).

To understand why CASCADE performs well, in Figure 8 we plot trajectories from Montezuma's Revenge. We see that the inclusion of a diversity-inducing objective does indeed lead to more diverse behaviors in the environment, with each policy exploring different regions of the room. In contrast, for PP2E, agents 3,4,5,6,7,9 all exhibit very similar behavior, likely indicating that random initialization alone is not sufficient to induce diversity when all agents optimize for the same objective.

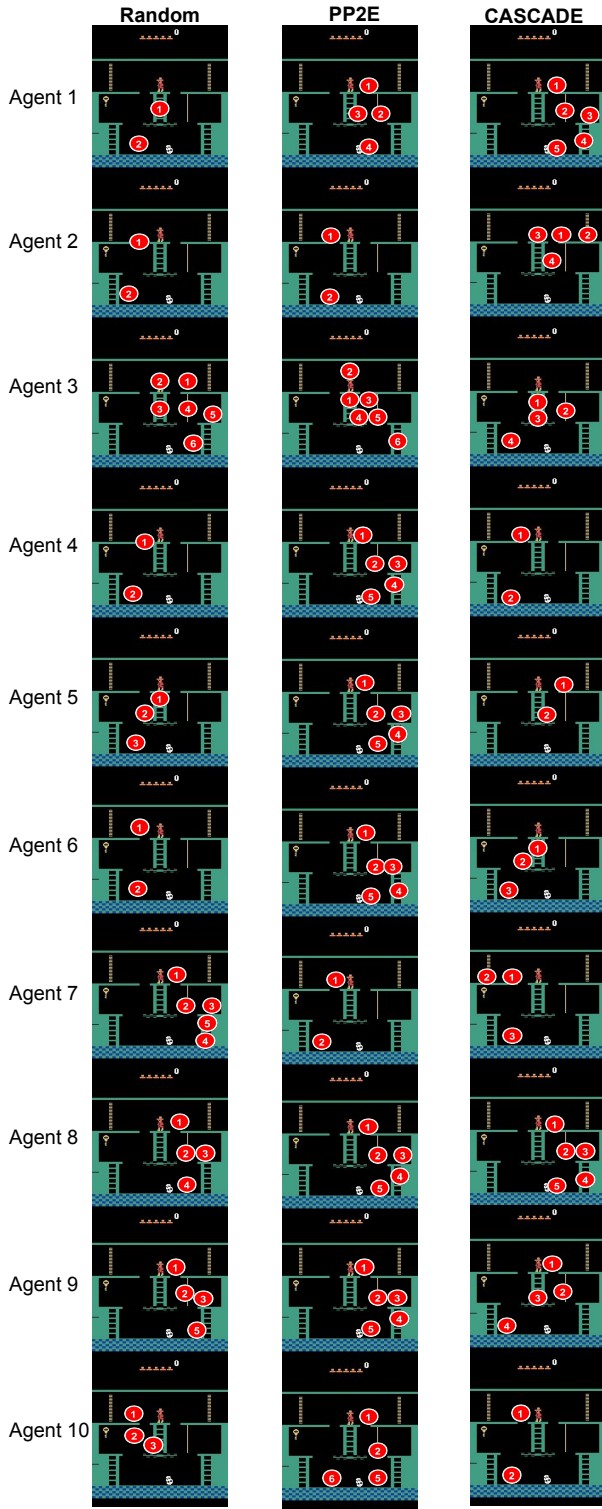

Figure 8: **Trajectories in Montezuma's Revenge**: Each row shows a trajectory from one of the 10 exploration agents of a model after 3M training timesteps (P2E is omitted because it only has 1 agent.) We can see that PP2E's agents exhibit the most homogeneous behaviors, and result in trajectories that focus on the bottom right of the room, while CASCADE agents manage to explore more of the room collectively.

## B Implementation Details

DreamerV2 consists of an image encoder that uses a Convolutional Neural Network (CNN, [56]), a Recurrent State-Space Model (RSSM [38]) that learns the dynamics, and predictors for the image, reward, and discount factor. The RSSM uses a sequence of deterministic recurrent states $h_t$. At each step, it computes a posterior state $z_t$ conditioned on the current image $x_t$, as well as a prior state $\hat{z}_t$ without the current image. During world model training, the concatenation of $h_t$ and $z_t$ is used to reconstruct the current image $x_t$, and predict the reward $r_t$ and discount factor $\gamma_t$. Once the world model is trained, it can be used to roll out "imaginary" trajectories where the model state is instead the concatenation of the deterministic state $h_t$ and prior stochastic state $\hat{z}_t$.

$$\text{RSSM} \begin{cases} \text{Recurrent model:} & h_t = f_\psi(h_{t-1}, z_{t-1}, a_{t-1}) \\ \text{Representation model:} & z_t \sim q_\psi(z_t | h_t, x_t) \\ \text{Transition predictor:} & \hat{z}_t \sim p_\psi(\hat{z}_t | h_t) \end{cases} \tag{7}$$

$$\begin{aligned} \text{Image predictor:} & \quad \hat{x}_t \sim p_\psi(\hat{x}_t | h_t, z_t) \\ \text{Reward predictor:} & \quad \hat{r}_t \sim p_\psi(\hat{r}_t | h_t, z_t) \\ \text{Discount predictor:} & \quad \hat{\gamma}_t \sim p_\psi(\hat{\gamma}_t | h_t, z_t). \end{aligned} \tag{8}$$

Our implementation (https://github.com/ainomearod/divwm) was built on top of the official DreamerV2 repository https://github.com/danijar/dreamerv2. We used the default hyper-parameter values in the DreamerV2 repository. Table 1 lists the additional hyperparameters used in our experiments.

We also incorporate a latent disagreement ensemble, following Plan2Explore [95]. This involves training, alongside the RSSM model, an MLP ensemble with 10 members (each having 4 hidden layers, 400 units per layer, and different parameter initializations). These one-step models take action $a_t$ and latents $z_t$, $h_t$ as inputs, and try to predict the next stochastic latent $z_{t+1}$. The variance across these outputs during imagined rollouts is then used as the reward signal that forms the Information Gain (InfoGain) objective. Note that this ensemble is otherwise unused, and is solely trained for the purposes of the exploration objective.

Table 1: CASCADE hyperparameters

| Environment | Parameter | Value |
|---|---|---|
| MiniGrid-FourRooms | CASCADE weight ($\lambda$) | 0.1 |
| | batch size | 200k |
| | deployments | 5 |
| | explorer train steps | 10k |
| | offline model train steps | 20k |
| MiniGrid-MultiRoom-N4S5 | CASCADE weight ($\lambda$) | 0.3 |
| | batch size | 200k |
| | deployments | 5 |
| | explorer train steps | 10k |
| | offline model train steps | 20k |
| Montezuma's Revenge | CASCADE weight ($\lambda$) | 0.8 |
| | batch size | 200k |
| | deployments | 15 |
| | explorer train steps | 10k |
| | offline model train steps | 5k |
| Walker | CASCADE weight($\lambda$) | 0.1 (Random) 0.3 (Medium/Expert) |
| | batch size | 200k |
| | deployments | 2 |
| | explorer train steps | 250 |
| | offline model train steps | 5k |

 # C  Deriving the Objective

 ## C.1  An Information Theoretic Perspective

Inspired by [95], we derive a data acquisition objective based on the mutual information between the collected data and parameters of the world model $\mathcal{M}_\psi$. $\mathcal{M}_\psi$ can be either stochastic or deterministic, and can also be an ensemble of empirical models. Crucially however, we eschew the reliance on one-step information gain (as performed in [95]), and instead aim to maximize diversity directly in the space of trajectories. To do this, let $\Phi : \Gamma \to \Omega$ be an summary function mapping trajectories into an embedding 'summary' space. For any model $\mathcal{M}_\psi$ we use the notation $\mathbb{P}_\pi[\mathcal{M}_\psi]$ to denote the distribution over trajectories generated by policy $\pi$ in $\mathcal{M}_\psi$ and use $\mathcal{M}_\psi \sim \mathbb{P}_{\mathcal{M}_\psi}$ to denote the posterior distribution over models. The later depends on the data collected so far.

In this work we consider different embedding functions $\Phi$,

1. **Final State Embedding**. In this setting we define $\Phi(\tau) = h_H$ where $h_H$ corresponds to the $H-$th (and last) state embedding in the trajectory $\tau$. Since we use an RNN world model, the use of this summary embedding is analogous to the final encoder embedding in seq2seq language models [101].

2. **Visitation Frequencies**. In the tabular setting if we define $\Phi(\tau)$ as a discounted count statistic for states in $\tau$, we recover $\mathbb{P}_\pi^\Phi[\mathcal{M}_\psi] = d_{\mathcal{M}_\psi}^\pi$ where $d_{\mathcal{M}_\psi}^\pi$ corresponds to the discounted visitation distribution of policy $\pi$ in model $\mathcal{M}_\psi$.

The authors of [95] study a per state-action mutual information objective that informs the construction of a greedy mutual information maximizing policy. Implicitly, this per-step objective assumes the model to factor in *independent* transition operators pertaining to each state-action pair. This is certainly not the case when using powerful function approximators such as neural networks. In this case, the dynamics model corresponding to the state transitions of a specific state action pair is correlated with the state transition model of *other* state-actions. This is not captured accurately by the objective in [95]. Instead, we consider the following choice for an exploratory policy in the single policy setting studied in [95]:

$$\pi_{\text{EXP}} = \max_\pi \mathcal{I}\left(\mathbb{P}_\pi^\Phi[\mathcal{M}_\psi]; \mathcal{M}_\psi\right) = \mathcal{H}(\mathbb{P}_\pi^\Phi[\mathcal{M}_\psi]) - \mathcal{H}(\mathbb{P}_\pi^\Phi[\mathcal{M}_\psi]|\mathcal{M}_\psi). \tag{9}$$

$\pi_{\text{EXP}}$ is a policy whose distribution over summaries $\Phi(\tau)$ with $\tau \sim \mathbb{P}_\pi[\mathcal{M}_\psi]$ and $\mathcal{M}_\psi \sim \mathbb{P}_{\mathcal{M}_\psi}$ has large entropy, but such that the average entropy of the summaries in every individual model is small. The term $\mathcal{H}(\mathbb{P}_\pi^\Phi[\mathcal{M}_\psi])$ captures the total uncertainty (epistemic + aleatoric) while subtracting the conditional entropy removes the aleatoric uncertainty resulting from noise within the model and the policy.

To gain intuition about this objective, consider the case where $\Phi$ equals the *final state embedding*. For simplicity, we consider the scenario when all models $\mathcal{M}_\psi \sim \mathbb{P}$ and all policies $\pi \sim \Pi$ are such that for any realization of $\mathcal{M}_\psi = m_\psi$, the conditional entropy $\mathcal{H}(\mathbb{P}_\pi^\Phi[\mathcal{M}_\psi]|\mathcal{M}_\psi = m_\psi) = \sigma(m_\psi)$ is a function of the world $m_\psi$ and not of the policy. In this case, the policy $\pi_{\text{EXP}}$ is the entropy maximizing policy:

$$\pi_{\text{EXP}} = \max_\pi \mathcal{H}(\mathbb{P}_\pi^\Phi[\mathcal{M}_\psi]).$$

The scenario above is realized for example when all models $\mathcal{M}_\psi \sim \mathbb{P}_{\mathcal{M}_\psi}$ are deterministic and all policies in $\Pi$ are deterministic. In this case, for every realization of $\mathcal{M}_\psi = m_\psi$, the conditional entropy $\mathcal{H}(\mathbb{P}_\pi^\Phi[\mathcal{M}_\psi]|\mathcal{M}_\psi = m_\psi) = 0$. The assumption $\mathcal{H}(\mathbb{P}_\pi^\Phi[\mathcal{M}_\psi]|\mathcal{M}_\psi = m_\psi) = \sigma(w)$ is also realized when the distribution $\Phi(\tau) \sim \mathbb{P}_\pi^\Phi[m_\psi]$ is approximated as a Gaussian distribution $\mathbb{P}_\pi^\Phi[m_\psi] \approx \mathcal{N}(\mu(m_\psi, \pi), \Sigma(m_\psi))$ with policy dependent mean $\mu(m_\psi, \pi)$ and policy independent covariance $\Sigma(m_\psi)$.

In these cases $\pi_{\text{EXP}}$ is the policy that produces the most diverse distribution over final states across the posterior distribution over models $\mathbb{P}_{\mathcal{M}_\psi}$. Since access to $\mathcal{H}(\mathbb{P}_\pi^\Phi[\mathcal{M}_\psi])$ may not be possible, in our experiments we make further approximations inspired by [95]. Under the same gaussian approximation $\mathbb{P}_\pi^\Phi[m_\psi] \approx \mathcal{N}(\mu(m_\psi, \pi), \Sigma(m_\psi))$, optimizing $\max_\pi \mathcal{H}(\mathbb{P}_\pi^\Phi[\mathcal{M}_\psi])$ is achieved by

finding the policy $\pi$ that makes the ensemble means $\mu(m_\psi, \pi)$ as far apart as possible. A suitable surrogate for this objective is to maximize the empirical variance over means,

$$\text{Var}(\pi) = \frac{1}{|\mathcal{M}_\psi| - 1} \sum_{m_\psi} \|\mu(m_\psi, \pi) - \mu'(\pi)\|^2$$

where $|\mathcal{M}_\psi|$ denotes the number of samples $m_\psi$, and $\mu'(\pi) = \frac{1}{|\mathcal{M}_\psi|} \sum_{m_\psi} \mu(m_\psi, \pi)$. We now consider a "batch" version of Eq. 9 involving a population of $B$ agents:

$$\{\pi_{\text{EXP}}^{(i)}\}_{i=1}^B = \max_{\pi^{(1)}, \cdots, \pi^{(B)} \in \Pi^B} \mathcal{I}\left(\prod_{i=1}^B \mathbb{P}_{\pi^{(i)}}^\Phi[\mathcal{M}_\psi]; \mathcal{M}_\psi\right) =$$

$$\mathcal{H}\left(\prod_{i=1}^B \mathbb{P}_{\pi^{(i)}}^\Phi[\mathcal{M}_\psi]\right) - \mathcal{H}\left(\prod_{i=1}^B \mathbb{P}_{\pi^{(i)}}^\Phi[\mathcal{M}_\psi]\Big|\mathcal{M}_\psi\right) \quad (10)$$

where $\prod_{i=1}^B \mathbb{P}_{\pi^{(i)}}^\Phi[\mathcal{M}_\psi]$ is the product measure of the candidate policies embedding distributions over the model $\mathcal{M}_\psi$. By definition the conditional entropy factors as,

$$\mathcal{H}\left(\prod_{i=1}^B \mathbb{P}_{\pi^{(i)}}^\Phi[\mathcal{M}_\psi]\Big|\mathcal{M}_\psi\right) = \sum_{i=1}^B \mathcal{H}\left(\mathbb{P}_{\pi^{(i)}}^\Phi[\mathcal{M}_\psi]\Big|\mathcal{M}_\psi\right). \quad (11)$$

The objective in Eq. 10 has a similar interpretation as in the single policy case. We are hoping to find a set of policies whose average conditional entropies are small (Eq. 4), but whose total entropies are large. Intuitively, we see this objective is more amenable to our population of policies. Concretely, by considering distributions directly over the space of trajectories, we ensure that each agent does not 'double count' the explored states under Eq. 10, whereas applying the same principal to a one-step information gain objective would simply ensure diversity *conditioned* on that state and action, and doesn't explicitly ensure diversity in the visited states by the population.

### C.1.1  Proof of Lemma 1

In the proof of Lemma 1 we will make use of the following supporting result,

**Lemma 3.** *Let $p \in (0, 1]$ and $p_1, p_2 > 0$ satisfying $p_1 + p_2 = p$ then,*

$$p \log(1/p) < p_1 \log(1/p_1) + p_2 \log(1/p_2).$$

*Proof.* Let $p_1 = \alpha p$ with $\alpha \neq 0, 1$, then,

$$p_1 \log(1/p_1) + p_2 \log(1/p_2) = \alpha p \log(1/(\alpha p)) + (1 - \alpha)p \log(1/((1 - \alpha)p))$$

$$= p \log(1/p) + p \left(\underbrace{\alpha \log(1/\alpha) + (1 - \alpha) \log(1/(1 - \alpha))}_{>0}\right)$$

$$> p \log(1/p).$$

$\square$

Lemma 3 implies the following result,

**Lemma 4.** *Let $\mathbf{p} \in [0, 1]^K$ be a vector satisfying $\sum_{i=1}^K \mathbf{p}_i = 1$ and let $\mathcal{C}$ be a partition of $[K]$ such that $|\mathcal{C}| \leq K - 1$. For all $C \in \mathcal{C}$ denote $\mathbf{p}(C) = \sum_{i \in C} \mathbf{p}_i$. The following inequality holds,*

$$\sum_{C \in \mathcal{C}} \mathbf{p}(C) \log(1/\mathbf{p}(C)) \leq \sum_{i=1}^K \mathbf{p}_i \log(1/\mathbf{p}_i).$$

If $\mathbf{p}_i > 0$ *the inequality is strict,*

$$\sum_{C \in \mathcal{C}} \mathbf{p}(C) \log(1/\mathbf{p}(C)) < \sum_{i=1}^{K} \mathbf{p}_i \log(1/\mathbf{p}_i).$$

*Proof.* W.l.o.g we call $C_1$ the partition set containing 1 and assume $\mathbf{p}_1 > 0$. If $\mathbf{p}_i > 0$ for only one element of $C_1$, it must be the case that $\mathbf{p}_1 \log(1/\mathbf{p}_1) = \mathbf{p}_{C_1} \log(1/\mathbf{p}_{C_1})$.

We now assume that $|C_1| > 1$ and that $C_1$ has at least two indices $i \neq j$ such that $\mathbf{p}_i, \mathbf{p}_j > 0$. Let $l$ denote the size of $C_1$ and w.l.o.g. let's say $C_1 = \{1, 2, \cdots, l\}$. Lemma 3 implies the following inequalities,

$$\mathbf{p}(C_1) \log(1/\mathbf{p}(C_1)) < \mathbf{p}_1 \log(1/\mathbf{p}_1) + \mathbf{p}(C_1 \backslash \{1\}) \log(1/\mathbf{p}(C_1 \backslash \{1\}))$$
$$< \cdots$$
$$< \sum_{i \in C_1} \mathbf{p}_i \log(1/p_i).$$

Applying this reasoning to each element in the partition $\mathcal{C}$ yields the desired result.

$\square$

**Lemma 1.** *When all models $\mathcal{M}_\psi$ in the support of the model posterior are deterministic and tabular, and the space of policies $\Pi$ consists only of deterministic policies, there always exists a solution $\{\pi_{\mathrm{EXP}}^{(i)}\}_{i=1}^{B}$ satisfying $\pi_{\mathrm{EXP}}^{(i)} \neq \pi_{\mathrm{EXP}}^{(j)} \forall i \neq j$. Moreover, there exists a family of tabular MDP models, such that the maximum cannot be achieved by setting $\pi_{\mathrm{EXP}}(i) = \pi$ for a fixed $\pi$.*

*Proof.* First let's observe that when the models and policies are all deterministic, the conditional entropies satisfy,

$$\mathcal{H}\left(\prod_{i=1}^{B} \mathbb{P}_{\pi^{(i)}}^{\Phi}[\mathcal{M}_\psi] \Big| \mathcal{M}_\psi\right) = \sum_{i=1}^{B} \mathcal{H}\left(\mathbb{P}_{\pi^{(i)}}^{\Phi}[\mathcal{M}_\psi] \Big| \mathcal{M}_\psi\right) = 0.$$

We assume the set of policies $\Pi$ is of size at least $B$. Otherwise, the result cannot be true. The first result follows immediately by Lemma 4. For any fixed $\pi$, the product distribution $\prod_{i=1}^{B} \mathbb{P}_{\pi}^{\Phi}[\mathcal{M}_\psi]$ is a distribution over 'diagonal' tuples of the form $\underbrace{(b, \cdots, b)}_{\text{size } B}$ where $b$ is an embedding.

Consider an arbitrary set of policies $\pi^{(2)}, \cdots, \pi^{(B)}$ satisfying $\pi^{(i)} \neq \pi^{(j)} \neq \pi$ for all $i, j \in \{2, \cdots, K\}$. Call $\pi^{(1)} = \pi$. The distribution over embeddings induced by the product distribution $\prod_{i=1}^{B} \mathbb{P}_{\pi^{(i)}}^{\Phi}[\mathcal{M}_\psi]$ is made of tuples of the form $(b_1, \cdots, b_B)$. Call $\mathbf{p}(b_1, \cdots, b_B)$ the probability under the product measure $\prod_{i=1}^{B} \mathbb{P}_{\pi^{(i)}}^{\Phi}[\mathcal{M}_\psi]$ of the tuple $(b_1, \cdots, b_B)$. Notice the 'projection measure' onto the first coordinate satisfies

$$\prod_{i=1}^{B} \mathbb{P}_{\pi^{(i)}}^{\Phi}[\mathcal{M}_\psi](b_1) = \prod_{i=1}^{B} \mathbb{P}_{\pi}^{\Phi}[\mathcal{M}_\psi](b_1, \cdots, b_1) := \mathbf{p}_1(b_1)$$

and that $\sum_{b_2, \cdots, b_K} \mathbf{p}(b_1, b_2, \cdots, b_K) = \mathbf{p}_1(b_1)$. This induces a partition over the outcome space $(b_1, \cdots, b_K)$ corresponding to $C(b_1) = \{(b_1, b_2, \cdots, b_K)\}_{b_2, \cdots, b_K}$.

A direct use of Lemma 4 implies the entropy of the product distribution over finer grained tuples is larger than the entropy over the diagonal of the product distribution having all coordinates equal to each other. In fact this result also informs when the inequality will be strict. This happens when (for example the measure over $\mathcal{M}_\psi$ is the counting measure) there exists two worlds $\mathcal{M}_\psi^{(1)}$ and $\mathcal{M}_\psi^{(2)}$ such that the embedding tuples $(b_1^{(1)}, \cdots, b_B^{(1)})$ and $(b_1^{(2)}, \cdots, b_B^{(2)})$ satisfy $b_1^{(1)} = b_1^{(2)}$ and $(b_2^{(1)}, \cdots, b_B^{(1)}) \neq (b_2^{(2)}, \cdots, b_B^{(B)})$.

We will use this observation to prove the second claim. Consider a family of MDPs formed of depth $L$ binary trees. In this family, the paths leading to the $L-1$ layer are the same, but the connections from layer $L-1$ to layer $L$ are unknown. The 'posterior' distribution is assumed to be uniform over all plausible trees. Layer $L-1$ has size $2^{L-1}$ and layer $L$ has size $2^L$. W.l.o.g. we assume the set of nodes in layer $L$ is labeled as $[1, 2, \cdots, 2^L]$. The distribution over models is supported over the set of partitions of size $2^{L-1}$ of $[1, \cdots, 2^L]$ where each partition set has size 2. We assume $2^L \geq B$ so that there are at least $B$ distinct policies.

Let $\pi$ be a fixed policy. It is enough to show there exist two worlds $\mathcal{M}_\psi^{(1)}$ and $\mathcal{M}_\psi^{(2)}$ such that policy $\pi$ ends in the same state for both $\mathcal{M}_\psi^{(1)}$ and $\mathcal{M}_\psi^{(2)}$ but that there exist distinct policies $\pi^{(2)}, \cdots, \pi^{(B)}$ such that their endpoints $(b_2^{(1)}, \cdots, b_B^{(1)})$ and $(b_2^{(2)}, \cdots, b_B^{(2)})$ in worlds $\mathcal{M}_\psi^{(1)}$ and $\mathcal{M}_\psi^{(2)}$ satisfy $(b_2^{(1)}, \cdots, b_B^{(1)}) \neq (b_2^{(2)}, \cdots, b_B^{(2)})$. The latter always holds because among the set of models that maintain the same endpoint for $\pi$, there is a pair of models that has different endpoints for $\pi^{(2)}$ for any arbitrary $\pi^{(2)} \neq \pi$. This suffices to show the entropy of the ensemble of distinct policies is strictly larger than the entropy of the 'diagonal choice' $\underbrace{(\pi, \cdots, \pi)}_{\text{size } B}$.

$\square$

### C.1.2 InfoCascade

Since the mutual information objective in Eq. 3 is submodular, a simple greedy algorithm yields a $(1 - \frac{1}{e})$ approximation of the optimum [72]. This is the same observation that gives rise to the greedy algorithm underlying other batch exploration objectives, such as in BatchBALD [51].

Let's start by assuming we have candidate policies $\pi^{(1)}, \cdots, \pi^{(i-1)}$. InfoCascade selects $\pi^{(i)}$ based on the following greedy objective,

$$\pi^{(i)} = \arg \max_{\tilde{\pi}^{(i)} \in \Pi} \mathcal{I}\left(\prod_{j=1}^{i} \mathbb{P}_{\tilde{\pi}^{(j)}}^{\Phi}[\mathcal{M}_\psi]; \mathcal{M}_\psi \Big| \tilde{\pi}^{(j)} = \pi^{(j)} \; \forall j \leq i - 1\right)$$

$$= \mathcal{H}\left(\prod_{j=1}^{i} \mathbb{P}_{\tilde{\pi}^{(j)}}^{\Phi}[\mathcal{M}_\psi] \Big| \tilde{\pi}^{(j)} = \pi^{(j)} \; \forall j \leq i - 1\right) - \mathcal{H}\left(\prod_{j=1}^{i} \mathbb{P}_{\pi^{(j)}}^{\Phi}[\mathcal{M}_\psi] \Big| \mathcal{M}_\psi, \tilde{\pi}^{(j)} = \pi^{(j)} \; \forall j \leq i - 1\right).$$

Equation 4 implies

$$\mathcal{H}\left(\prod_{j=1}^{i} \mathbb{P}_{\pi^{(j)}}^{\Phi}[\mathcal{M}_\psi] \Big| \mathcal{M}_\psi,, \tilde{\pi}^{(j)} = \pi^{(j)} \; \forall j \leq i - 1\right) = \sum_{j=1}^{i} \mathcal{H}\left(\mathbb{P}_{\pi^{(i)}}^{\Phi}[\mathcal{M}_\psi] \Big| \mathcal{M}_\psi\right) \quad (12)$$

and therefore if we approximate $\mathbb{P}_\pi^{\Phi}[m_\psi]$ by a Gaussian $\mathbb{P}_\pi^{\Phi}[m_\psi] \approx \mathcal{N}(\mu(m_\psi, \pi), \Sigma(m_\psi))$, the conditional entropy becomes a policy independent term. In this case finding $\pi^{(i)}$ boils down to solving for the policy that maximizes $\mathcal{H}\left(\prod_{j=1}^{i} \mathbb{P}_{\pi^{(j)}}^{\Phi}[\mathcal{M}_\psi] \Big| \pi^{(1)}, \cdots, \pi^{(i-1)}\right)$. Using the same approximations described for the single policy objective, a suitable surrogate for this objective is to maximize $\max_\pi \text{Var}(\pi|\pi^{(1)}, \cdots, \pi^{(i-1)})$, the empirical variance over means with respect to the policies found so far $\{\pi^{(j)}\}_{j=1}^{i-1}$,

$$\text{Var}(\pi|\pi^{(1)}, \cdots, \pi^{(i-1)}) =$$
$$\frac{1}{|\mathcal{M}_\psi||i| - 1} \sum_{m_\psi} \sum_{\tilde{\pi} \in \{\pi^{(1)}, \cdots, \pi^{(i-1)}, \pi\}} \|\mu(m_\psi, \tilde{\pi}) - \mu'(\pi, \pi^{(1)}, \cdots, \pi^{(i-1)})\|^2$$

Where $\mu'(\pi, \pi^{(1)}, \cdots, \pi^{(i-1)}) = \frac{1}{|\mathcal{M}_\psi||i|} \sum_{m_\psi} \sum_{\tilde{\pi} \in \{\pi^{(1)}, \cdots, \pi^{(i-1)}, \pi\}} \mu(m_\psi, \tilde{\pi})$.

# D  Theoretical RL Intuition

The problem we study in this work can be thought of as the "batch" version of the Reward Free Exploration formalism [44]. In this setting, the learner interacts with an MDP in two phases: 1) a training phase, where it is allowed to collect data from the environment; 2) a test phase, where the learner is presented with an arbitrary task (parametrized by a reward function unseen during training) and it must produce a near optimal policy. When faced with this problem, the learner is required to design a careful exploration strategy that permits them to build an accurate model in all areas of the state-space that can be reached with sufficient probability. There have been multiple follow up works that have also studied this problem in the context of Linear MDPs [113] and neural function approximation [85]. Nonetheless, there has been limited focus on the batch setting, where the learner is required to collect data via parallel data gathering policies in each training phase.

Due to the adaptive nature of the algorithm (every single environment query uses all information collected so far), there is an inevitable drop in performance when moving from the sequential ( fully online) setting to the batch (deployment efficient) setting, as measured by the total number of environment interactions (in our case, $T \times B$ where $T$ is the number of training rounds and $B$ the number of parallel collection policies). To illustrate how CASCADE mitigates this loss in sample-efficiency, we outline the pseudo-code of CASCADE-TS, a greedy Thompson Sampling algorithm (see [4]) that produces the $i-$th batch exploration policy in a tabular enviornment by incorporating fake count data from rolling out policies $\pi^{(1)}, \cdots, \pi^{(i-1)}$ in the model. In the algorithms we present in the main body, we incorporate data gathered in the model by previous $i-1$ selected policies into the batch when building the $i-$th exploration policy; this is directly related to the approach behind CASCADE-TS. Concretely, encouraging policy $\pi^{(i)}$ to induce a high variance among the embeddings produced by $\{\pi^{(1)}, \cdots, \pi^{(i-1)}\}$ can be thought as adding a bonus to encourage $\pi^{(i)}$ to visit regions of the space that have a low embedding visitation count by the previous policies in $\{\pi^{(1)}, \cdots, \pi^{(i-1)}\}$.

---

**Algorithm 2** CASCADE-TS

1: **Input:** Exploration batch size $B$. Fake samples parameter $M$.
2: Initialize model $\hat{\mathbb{P}}_0$ over $\mathcal{S} \times \mathcal{A}$ state actions.
3: Initialize the batch collection policies as $\{\pi_0^{(i)}\}_{i=1}^B$ to uniform.
4: **for** time in $k = 0, 1, 2, \ldots$ **do**
5:     Collect trajectory data $\{\tau_k^{(i)}\}_{i=1}^B$ from $\pi_k^{(i)}$ for all $i = 1, \cdots, B$. Update $\mathcal{D}^{k+1} \leftarrow \mathcal{D}^k \cup \{\tau_k^{(i)}\}_{i=1}^B$.
6:     Initialize the fake data buffer $\mathcal{D}_{\text{fake}}^{k+1} = \emptyset$.
7:     **for** $i = 1, \cdots, B$ **do**
8:         Sample MDP model from posterior $\widetilde{\mathbb{P}}_{k+1}^{(i)} \sim \mathbb{P}(\cdot | \mathcal{D}^{k+1} \cup \mathcal{D}_{\text{fake}}^{k+1})$.
9:         Compute fake counts $N_{k+1}^{(i)}(s, a) = \sum_{(\tilde{s}, \tilde{a}, \tilde{s}') \in \mathcal{D}^{k+1} \cup \mathcal{D}_{\text{fake}}^{k+1}} \mathbf{1}(\tilde{s} = s, \tilde{a} = a)$.
10:        Compute fake uncertainty bonuses $b_{k+1}^{(i)}(s, a) \propto \sqrt{\frac{1}{N_{k+1}^{(i)}(s,a)}}$
11:        Solve for $\pi_{k+1}^{(i)}$ by solving,

$$\pi_{k+1}^{(i)} = \max_{\pi} \mathbb{E}_{\tau \sim \widetilde{\mathbb{P}}^{(i),\pi}_{k+1}} \left[ \sum_{h=1}^{H} b_{k+1}^{(i)}(s_h, a_h) \right]$$

12:        Collect fake trajectory data $\{\tau_{\ell,\text{fake}}^{(i)}\}_{\ell=1}^M$ from posterior sampled model $\widetilde{\mathbb{P}}_{k+1}^{(i)}$.
13:        Update fake data buffer $\mathcal{D}_{\text{fake}}^{k+1} \leftarrow \mathcal{D}_{\text{fake}}^{k+1} \cup \{\tau_{\ell,\text{fake}}^{(i)}\}_{\ell=1}^M$.
14:    **end for**
15: **end for**

---

Algorithm 2 works by sampling a model from a model posterior, solving for an optimal uncertainty seeking policy and updating the model with 'fake' data corresponding to trajectories collected in the model from this uncertainty seeking policy. After producing the $i-$th exploration policy in the batch, the recomputed uncertainty bonuses of the model are reduced in the areas of the state space

most visited by the $i-$th policy. This will encourage subsequent uncertainty seeking policies (i.e., $\pi_{k+1}^{(i+1)}, \pi_{k+1}^{(i+2)}, \ldots, \pi_{k+1}^{(B)}$) to visit parts of the space not yet explored by previous policies.

**Why Thompson Sampling?** The reader may wonder why Algorithm 2 samples a model $\widetilde{\mathbb{P}}_{k+1}^{(i)}$ from a TS posterior instead of using the empirical model resulting from fake and true data. The answer is that the raw empirical model may ascribe a probability of zero to certain transitions, which means we may miss exploring parts of the true state-action space. Having a TS prior that ascribes non-zero probabilities to all possible transitions fixes this potential issue.

### D.1   Proof of Lemma 2

Let's consider the class of deterministic MDPs with $S = |\mathcal{S}|$ states and $A = |\mathcal{A}|$ actions. It is clear that playing the same deterministic policy multiple times does not provide us with any more information than playing it once. We will then show that running Thompson TS, the learner will end up with a nonzero probability of producing at least two distinct policies within a batch of size $B > 2$.

We will further assume the posterior is aware of the deterministic nature of the MDP family so that any sample MDP model $\widetilde{\mathbb{P}}_{k+1}^{(i)}$ is deterministic. For $i > 1$, the model $\widetilde{\mathbb{P}}_{k+1}^{(i)}$ is one whose transitions are consistent with the fake data generated by $\pi_{k+1}^{(1)}, \cdots, \pi_{k+1}^{(i-1)}$ (and the true data in $\mathcal{D}^k$).

A deterministic MDPs can be encoded as a set of triplets $(s, a, s')$. We say a model $\{(s, a, s')\}_{s \in \mathcal{S}, a \in \mathcal{A}}$ is $\epsilon-$accurate if at least a fraction of $1 - \epsilon$ of the triplets are correct.

We now illustrate how the CASCADE-TS algorithm evolves in this setting. To simplify our task, we will further restrict our attention to the family of deterministic MDPs made of binary trees of height $L$. The number of nodes in any of such trees equals $2^{L+1} - 1$. We will define the counts to be a large value $(2L)$ when no data has been collected of a given state action pair. When data has been collected, we define the bonus terms to be of order at most $1$.

To figure out the connectivity of any such trees it is necessary to try out $2^L - 1$ distinct sequences of $L$ actions (the nature and connectivity of the remaining leaf nodes can be inferred once all the other ones have been decoded). To build an $\epsilon-$accurate model, it is enough to know the connectivity structure of $1 - \epsilon$ proportion of paths corresponding to $(1 - \epsilon)(2^L - 1)$ leaf nodes.

To prove the result of Lemma 2, we observe that any batch strategy can be simulated by a fully sequential learner, so it immediately follows that $T(\epsilon, \text{Sequential}) \leq T(\epsilon, \text{AnyBatchStrategy})$. To show that CASCADE-TS will be better than SinglePolicyBatch in this tree example, as long as $B$ is smaller than the remaining number of paths the learner has not tried out, it will produce a set of $B$ policies different from any policy played so far and also different from each other.

To see why the second inequality is true, it is enough to see that complete paths (real or imagined) have a total reward score of at most $L$, since the counts are always at most one for each edge that is present. Nonetheless, the counts over any path (sequence of $L$ actions) that has not been visited nor has been imagined to be visited will be at least $2L$. Thus, every step in the sequential batch construction mechanism will produce a new policy (sequence of $L$ actions) not existent neither in the true nor the fake data so far. This finalizes the proof. Notice that the SinglePolicyBatch strategy will have tried only $K$ unique paths after $K$ batches have been collected, whereas the CASCADE-TS strategy will have tried $BK$, resulting in a potentially substantial improvement in sample efficiency.