# OpenReview forum: "Learning General World Models in a Handful of Reward-Free Deployments"
_NeurIPS.cc/2022/Conference — NeurIPS 2022 Accept_

### Official Review · Reviewer_bwKR · 2022-07-06

**Rating:** 5
**Confidence:** 4
**Soundness:** 3 good
**Presentation:** 2 fair
**Contribution:** 2 fair

**Summary:**

The paper introduces a setting called reward-free deployment efficient setting, where one can interact with the environment with limited number of times/limited number of updates (but could with multiple agents in parallel), in the reward-free fashion. The paper proposes to solve this problem by learning a world model with a network ensemble, and with a certain number of exploration policies. The paper presents to train the world model in a similar fashion as DreamerV2, and train the exploration policies with a composition of heuristic objectives. Finally, the paper evaluates the effectiveness of the algorithm on both game and locomotion benchmarks.

**Questions:**

## Major Questions

1. In Figure 1, the top two rows seem identical. Thus there are a lot of redundancies in the figure and in fact seems a little bit strange. Also, the definition of deployment seems to be the number of interactions that the algorithm itself will have with the environment? Thus the number of deployments of offline RL should be 0? Otherwise, if the offline dataset is collected from a heterogeneous source, the number of deployments of offline RL could also be large?

2. The definition of contextual mdp (CMDP): according to [3], the definition of CMDP seems to be that we have a set of mdps, where each mdp, in addition to have different reward, but also have different dynamics conditioned on the context?

3. The derivation of the final objective is very confusing. In the main text, the ImagDiv term seems to be a heuristic surrogate of the first entropy term in line 140. Under the assumption that the conditional latent distribution is a gaussian with variance unrelated to the policy, which seems to be the assumption made for section 3.2, the ImagDiv objective is actually estimating the whole objective? However, in the appendix, the paper also presents the derivation for the InfoGain term, under the same assumption. This is confusing because the paper tries to use two different objectives to estimate the same thing?

4. In the definition of ImageDiv term, is the expectation also over the model ensembles? Also $\Phi$ is not defined in the ImageDiv term but is defined in the appendix for the InfoGain term. Is $\Phi$ in ImagDiv the same as $\mu(m_{\psi}\pi)$?

5. In Fig.3, why is PP2E10's zero-shot success rate higher in a more challenging environment?

6. In the experiment for locomotion, why do we need to assume that we have access to the offline dataset? It is known that offline RL needs good coverage from the offline dataset to perform well, thus it looks like in this setting the algorithm already gets a good warm-start? How would the method perform without the offline dataset?

### references

[3] Hallak, Assaf, Dotan Di Castro, and Shie Mannor. "Contextual markov decision processes." arXiv preprint arXiv:1502.02259 (2015).

## Minor issues

- In algorithm 1, it seems like the algorithm will never terminate?

- Many hyperlinks to the sections in the appendix are wrong. For example, line 136.

- Line 140, on the right hand side of the last equality, in the second entropy term, the distribution of latents should be conditioned on $\tilde \pi^{(j)}$'s but not $\pi^{(j)}$'s (the subscripts).

- The x-axis in Fig.5 is different from the text description.

**Limitations:**

1. The result in Lemma 1 assumes deterministic MDP and deterministic policy. Would it be obvious that one policy may not be able to explore the whole environment?

2. Section 3.3 seems to be in an awkward position. After a very heuristic description of the deep RL approach, section 3.3 returns to a theoretical analysis of a variant of the algorithm, and the setup is also simpler (tabular MDP). Maybe it could instead serve as a motivating section?

**Strengths And Weaknesses:**

## Strength

1. The idea of using a set of exploration policies is a crucial contribution, which also ensembles the idea of "policy cover" used for exploration in multiple theory papers. The paper provides some good intuition on why using beyond one fixed policy is important for exploration, such as Figure.2. However, the theory component does not seem very surprising, which will be more detailed explained in the latter part of the review.

2. The new reward-free deployment efficient setting indeed has some practical significance, especially in the robot learning scenario as the paper mentioned. The combination of reward-free exploration is also well justified. It would be more interesting to see if the proposed algorithm can indeed be adapted to the robot learning scenario, if possible.

3. The performance of the algorithm, especially on the atari game benchmark, is competitive. Although the setting of games may not be a very good fit for the deployment efficient setting, it also shows that the proposed algorithm can indeed be a good algorithm for just performing reward-free exploration. It would be interesting to see any comparison to algorithms with explicit exploration bonuses.

## Weakness

1. The paper is not very well-origanized. Some of the results are not crucial, and there are many unclear parts and unjustified components of the algorithm/derivation. Details will be listed in the questions and limitations section.

2.  The idea of learning a global accurate dynamics model, with a number of exploration policies based on the current dynamics models, in the reward-free exploration section, is not new (c.r. the idea of policy cover mentioned above). There are many theory papers that considered the similar scenario and proposed algorithms with similar intuition, for example, [1,2], which in fact undermines the novelty of the paper.

### references

[1] Agarwal, Alekh, et al. "Flambe: Structural complexity and representation learning of low rank mdps." Advances in neural information processing systems 33 (2020): 20095-20107.

[2] Modi, Aditya, et al. "Model-free representation learning and exploration in low-rank mdps." arXiv preprint arXiv:2102.07035 (2021).

---

> ### Author Response · Authors · 2022-08-02
> **Thank you for your review! [1/2]**
>
> Thank you for your review! We are pleased to see you appreciate that our work builds on theoretical foundations and applies them in a novel deep RL setting. We are optimistic that the issues raised in the review are things we can resolve over the coming days, and we hope you can subsequently consider supporting our paper’s acceptance. In particular, we have made a concerted effort to emphasize the existing theoretical work in this space. We appreciate that we previously did not include sufficient references here, but we do want to highlight that we put a tremendous effort into citing previous work (>100 citations in v1), so this was entirely unintentional. We hope that now with appropriate recognition, it is clear our work extends these ideas to the deep RL paradigm, which makes a solid contribution for NeurIPS.
>
> See below for our detailed responses to your Major Weaknesses.
>
> ### W1: The paper is not very well-origanized.
>
> We will seek to address these issues in the remaining responses.
>
> ### W2: Existing theory undermining novelty.
>
> We want to emphasize this work is not theoretical in nature. It is a methods paper to which we have added what we believe are useful theoretical intuitions. We also do not claim to have invented the theory of reward free exploration. In fact, we make that clear in the related work section (which has been expanded upon). In this work we consider a reward free setting where the learner can deploy exactly $B$ policies at each time-step. Our contribution is to adventure an answer and devise methods for this problem inspired by an information theoretic approach borrowed from existing work in active learning. We would be very excited if our work could spur a more thorough theoretical exploration of these ideas, of which (in the batch setting) *we are the first to port to deep RL*.
>
> We have added [1] and [2] to our related work section. Nonetheless we respectfully disagree with the reviewer in ascribing much more similarity between [1,2] and our work than sharing the objective to efficiently build a model by explicitly maximizing state coverage. First, we demonstrate our algorithms to be extremely effective (SOTA) in the neural network approximation regime, something that neither FLAMBE nor MOFFLE do as they only work for low rank MDPs, a regime that is far from describing many practical problems such as the large scale problems in our experiments. Second, we are chiefly invested in the concept of designing a diverse set of policies for exploration. This is an explicit objective of our work, and one that in practical settings we believe is crucial for the batch exploration problem we are proposing and one for which many theoretical works (the ones you cited included) do not provide an answer to.
>
> ### Q1: Figure 1
>
> The top row represents data collection and the second represents learning. The key idea here is learning occurs offline, rather than with some deep RL methods where the agent pauses every timestep to take a gradient update. The Figure was based on a similar diagram in [3] which introduced the deployment efficiency setting. To address the reviewer’s concern we have updated the Figure to make the agents distinct in the top two rows (representing deployment and learning). In [3] the offline setting is shown as 1, but we agree it makes more sense to give it a 0, so we also made that change. Thank you for the suggestions!
>
> ### Q2: Contextual MDP
>
> Indeed, it is entirely possible there are also different dynamics for each context. In this paper we consider the common case where the dynamics are the same across each context, but the reward function changes. We still find the CMDP formalism helpful, since it is being adopted more broadly in the community, but agree this special case is unclear in the paper. We have updated the manuscript to reflect this.
>
> ### Q3: Derivation of the objective/mismatch between main and Appendix
>
> We apologize for the confusion here, and note that the appendix *only* contains a derivation for the ImagDiv term. We hope that the general response has cleared up how the ImagDiv term and InfoGain are inspired by the core idea of maximizing mutual information, but differ considerably in the types of policies that they induce. Concretely, ImagDiv tries to ensure overall diversity across trajectories, whilst InfoGain tries to maximize the one-step epistemic uncertainty over imagined MDPs, which is less amenable when considering populations of agents. We will make sure this is communicated properly in the appendix.

---

> > ### Author Response · Authors · 2022-08-02
> > **[2/2]**
> >
> > ### Q4: Clarifications
> > See answer to Q3. In our experiments, $\phi$ is the last hidden state of a trajectory, but is generally a transformation that converts trajectories into embeddings. The expectation is over trajectories in the RSSM world model itself, which includes marginalizing over the stochastic latent variable.
> >
> > ### Q5: PP2E10 performance in Fig3
> >
> > In practice what makes an environment “easy” or “hard” for a given method is an active area of research [4], which is poorly understood in the deep RL community. In this case, both environments are procedurally generated so while FourRooms looks simple (and is often presented as a static grid world in theory papers) it is actually a non-trivial problem for deep RL agents. Note also the observation is pixel-like and it is partially observable and procedurally generated, so the agent may overfit to specific layouts explored better at train time.
> >
> > ### Q6: Offline Dataset
> >
> > This is a great question since we could certainly have clarified things further. The interesting thing here is we want to show that given *any* offline dataset, we can do subsequent deployments and build a more general model. The specific datasets used are challenging because generally models do not learn well from expert data in particular, since it is a narrow distribution, thus exploring online is paramount to ensuring adequate coverage in the collected data. Furthermore, the “random” datasets represents training from scratch with no prior knowledge, since any of these approaches could deploy a random policy for their first deployment. We have clarified this in the paper.
> >
> > ### Minor Issues
> >
> > Correct, the algorithm will never terminate. In theory it is open-ended the same as any other RL method. However, we understand this is in contrast to some of the motivational examples (e.g. deployment on robots) so we will clarify this to make the open-ended application more explicit.
> >
> > Other issues should all be fixed in the revised manuscript.
> >
> > Thank you again, we look forward to hearing from you in the coming days!
> >
> > [1] Agarwal, Alekh, et al. "Flambe: Structural complexity and representation learning of low rank mdps." Advances in neural information processing systems 33 (2020): 20095-20107.
> >
> > [2] Modi, Aditya, et al. "Model-free representation learning and exploration in low-rank mdps." arXiv preprint arXiv:2102.07035 (2021).
> >
> > [3] Matsushima et al., Deployment-Efficient Reinforcement Learning via Model-Based Offline Optimization, ICLR21
> >
> > [4] Furuta et al., Policy Information Capacity: Information-Theoretic Measure for Task Complexity in Deep Reinforcement Learning, ICML21

---

> > > ### Comment · Reviewer_bwKR · 2022-08-07
> > > **Responses**
> > >
> > > I would appreciate the authors for their detailed explanations. I would want to address two issues in my follow-up:
> > >
> > > 1. With the additional explanations, it seems like the major difference between the two terms, on a very high level, is "trajectory-level" vs. "point-wise" uncertainty (according to my understanding). However, it is still very unclear to me in the current form of the paper for the intuition of these two terms. My major confusion comes from: the two terms seem to both be surrogates for the same objective term (the info gain), and both of the derivations check out to me. Thus maybe I am missing something, but the current presentation looks still unclear to me which is the source of the difference between the two final surrogates.
> > >
> > > 2. Regarding the citation: I appreciate the authors' effort in being as inclusive as possible. However, my point of bringing up the theory paper is merely to show that reward-free exploration is a long-standing problem in theory and thus it is not a very novel setting so to speak. The papers I mentioned in my reviews are just very few selective papers on this topic and are just for the purpose of justifying points. I would also like to mention that, in my humble personal perspective, we should recognize the connections between theory and practical papers that study the same broad problems.
> > >
> > > In summary, I think the authors are making great efforts in improving their presentation on the paper. The paper has no clear technical issue and the empirical results are impressive. I would like to improve my rating by 1 but refrain from further improvement since my major confusion is still not completely address.

---

> > > > ### Author Response · Authors · 2022-08-08
> > > > **Thank you! and additional clarifications :)**
> > > >
> > > > Thank you for taking the time to read our response, and we are glad to have cleared up several details! We hope that we can further elucidate on the points you mention:
> > > >
> > > > 1. Both of these objectives (ImagDiva and InfoGain) can be derived from the principle of maximizing information gain, a technique that is much older than Plan2Explore and has its origins in the classical Bayesian Experiment Design Literature. It may aid the reviewer to know that our ensemble information objective can be (roughly speaking) thought of as a reinforcement learning version of the batch acquisition objective of BatchBALD (See equation [1]). The ImagDiv term is derived using an information gain objective over trajectory embeddings (these take into account the whole trajectory) and also take into account the $B$ parallel deployment agents. This is derived from trying to maximize an objective of the form $I( \Phi(\tau_1), \cdots, \Phi(\tau_B) \parallel W)$ where $W$ is the posterior over models and $\prod_{i=1}^B \Phi(\tau_i)$ is the product distribution over embeddings where $\tau_i$ is a trajectory sampled from policy $\pi_i$. The InfoGain objective on the other hand is the same term from Plan2Explore, and it is derived for each individual agent; in a population context this means that per-state they would take different actions, but this would not necessarily induce different trajectories. Moreover, the InfoGain objective is derived by maximizing an \textbf{expected} mutual information objective $\mathbb{E}_{(s,a) \sim \pi}[  I( h’ \parallel  W | s,a ) ] $ where $h’$ is the image embedding of the successor state to $s,a$ under policy $\pi$. These two terms are **not** the same. To use an example, consider an MDP where there are 3 actions being A, B and C, of which A and B lead to the same uncertain next state, and assume one agent in the population has already taken action A. Plan2Explore would prefer action B, as that generally reduces model uncertainty, whereas ImagDiv would prefer action C, as that action explicitly leads to a different state compared to previous agents, inducing “deeper” exploration. Concretely, ImagDiv is an explicit ensemble diversity inducing objective, while InfoGain is a per-state expectation of mutual information terms. However, we agree with the reviewer that the difference may not be immediately obvious, so also propose to rename InfoGain as LocalInfoGain in the CRC to make the difference clearer.
> > > > 2. We wholeheartedly agree with the reviewer. That is precisely why we called our paper ‘reward free’. We have not claimed to have invented reward free deployments, we have in fact brought this language into the applied community (recognizing this has been extensively studied in theoretical works) while also introducing a version of reward free exploration that has (to our knowledge) not been studied in theory, where we have a specific number (and a priori known) of reward free deployments per round $B$. We also think that our solution based on information gain would be an exciting avenue for theoretical researchers to delve into. The observation behind our work (and many others in parallel Bayesian optimization in supervised learning) that diversity aids in parallel exploration has not been fully characterized theoretically. We hope that our work can start this conversation.
> > > >
> > > > Again, we thank the reviewer for their time, and hope that our additional clarifications have helped address any major confusion that remains. If this is the case, then we kindly hope the reviewer provides additional support for our paper by increasing their score.

---

### Official Review · Reviewer_Yjhj · 2022-07-07

**Rating:** 5
**Confidence:** 4
**Soundness:** 3 good
**Presentation:** 1 poor
**Contribution:** 2 fair

**Summary:**

# New setting for reinforcement learning

The paper first introduces **reward-free deployment efficiency**, a new setting for reinforcement learning.

The *deployment efficiency* part means that we limit the number of agent versions that we deploy in the environment to collect new experience (for scalability reasons).

The *reward-free* part means that the collection procedure is task-agnostic. It aims at exploring the state-action space as much as possible without reward consideration (for generalization reasons).

# New algorithm within this setting

Within this setting, they propose **CASCADE**, an algorithm to deploy a fleet of *coordinated* agents that gather task-agnostic experience in the environment.

Similarly to [Plan2Explore](https://arxiv.org/pdf/2005.05960.pdf), the exploring behavior is trained with an intrinsic reward that estimates the uncertainty of a world model (they use DreamerV2 world model). This objective has an information-theoretic interpretation.  In the same way as Plan2Explore, they provide reward labels at test time, after data collection, to train a reward predictor for the world model. They can then train a task-specific agent in the imaginary MDP defined by the world model.

The major difference is to have different exploration policies and to *coordinate* them, so that they can organize together to explore different parts of the world.

They provide a theoretical motivation for this diversity of behaviors on a tabular version of their method (CASCADE-TS).

Their experiments demonstrate the superiority of CASCADE over random exploration and Plan2Explore, on MiniGrid, Montezuma's Revenge, and 4 tasks in DeepMind Control (stand, walk, run, flip).



**Questions:**

1) The `while True` in Algorithm 1 is slightly confusing since the deployment efficiency is precisely to limit the number of iterations of this loop, which here is infinite. Why couldn't you use a `for` loop with the number of deployments instead? That would allow to explicitly mention the number of deployments and precise that it is supposed to be small.

2) L97, could you explain why updating $\pi_{EXP}$ using the uncertainty of the imaginary MDP is different from Plan2Explore?

3) Figure 3, in FourRooms, how is it that P2E, with a lower state coverage and less rewarding episodes found, seems to beat PP2E on zero-shot success rate?

4) Still in Figure 3, could you explain more why CASCADE consistently reaches 100\% zero-shot success rate, while its number of rewarding episodes and state coverage are very similar to PP2E?

5) Figure 5, I would expect the 4 methods to be ranked randomly at deployment 0 (they all have access to the same data to start with), but across 30 seeds and 4 tasks, it looks like CASCADE is above the other ones even at deployment 0. Is there any reason for that?

6) Would it be possible to address the weaknesses pointed out in the "Strength And Weaknesses" sections? In particular, experiments on more Atari games / Crafter, comparison to Plan2Explore with their number of deployments and environments, and clarification of the formalism.


**Limitations:**

Section 4.3 clearly mentions some limitations. In particular, the authors make it clear that the learnt zero-shot policies are far from being optimal, which is understandable considering the difficulty of the reward-agnostic experience collection.

They do not discuss the potential negative impacts of their method, but clearly explain why the "deployment efficiency" constraint is useful in terms of security and cost.


**Strengths And Weaknesses:**

# Strengths

- Clear demonstration of the ability to explore the world in MiniGrid toy environments and Atari Montezuma's Revenge, with clear visualizations.

- Interesting improvements over an initial dataset of demonstrations with very few deployments in DeepMind Control.

- They provide a theoretical motivation behind the intuition that it is better to have diverse exploration policies.

---

# Weaknesses

## Lack of experimental evidence

The experiments about world exploration (4.1) are in toy environments of MiniGrid (FourRooms and MultiRoom) and in only one Atari game (Montezuma's Revenge). Running all Atari might be too much, but some Atari games present interesting world exploration problems (Hero, Freeway, Frostbite for instance).

Also, [Crafter](https://github.com/danijar/crafter) would be a much less trivial environment to benchmark the capabilities of CASCADE: procedurally generated world with many assets and interesting behaviors to discover.

---

## Deployment-efficiency limits comparison to prior work

The deployment-efficiency setting seems a bit artifical here. Indeed, it is justified in the text by scalability reasons, that are important when it comes to real world problems, but the environments considered in this work are classical toy environments.

Introducing a restriction on the number of deployments prevents proper comparison to prior work. In particular, it would have been really interesting to compare CASCADE with Plan2Explore (also reward-free / task-agnostic) by using their number of deployments and all of their DMC environments (a superset of the ones considered here).

Introducing this new setting (which is not justified by any real world environments) is risky since the comparison / novelty / interest are much harder to assess with a change in the experimental setup.

---

## The formalism is sometimes very unclear

In particular, L111 to L125, the information theoretic objective taken from Plan2Explore (Equation 1) is confusing.

First, looking at [Plan2Explore](https://arxiv.org/pdf/2005.05960.pdf):
- The intrinsic reward is to collect experience that maximizes the world model uncertainty, and they estimate this uncertainty via the empirical variance of an ensemble of next WM-latent predictors.
- They clearly state that this objective only *approximates* an information gain, and they frame it as a nice information theoretic interpretation of their objective.

Here, equation 1 states that the exploration policy is the argmax of the mutual information term, which is confusing since it is not directly optimized in Plan2Explore but approximated. Also, L116 mentions several MDPs to explain the mutual information term, who are they? In fact, in Plan2Explore, it is clear that there is a *single* imaginary MDP (defined by a single world model), and that they consider the disagreement in the ensemble of predictors, all trained to predict the dynamics of the same MDP dynamics. The explanation with several MDPs does not seem to make sense. This ensemble of predictors seems to be the central piece of the "information theoretic" objective, but is only mentioned in Appendix B as an implementation detail.

Also, in Plan2Explore, the mutual information is taken with a random variable $w$ that represent the optimal dynamics parameters, in a Bayesian way. Here, in equation 1, what is the mutual information between a distribution of states and an imaginary markov decision process?

---

## Theory is more a motivation than a guarantee

From what I understand, they prove that is is better to have diverse explorers (Lemma 1) and that CASCADE-TS (tabular version of CASCADE) learns faster or similarly to a naive fleet of identical explorers. However, there is no guarantee that CASCADE-TS actually converges to a solution of the optimization problem.

The name of section 3.3 is clear on that point ("Theoretical Motivation") but the abstract claim is not ("theoretical guarantees").

---

## The writing could be more polished

Some typos, and the notations could be more rigorous (e.g. L92 $\rho_0$ not defined, L137 $e$ is not defined).

---

# Originality

Originality is not clearly a strength or a weakness here. The work relies on the formalism of Plan2Explore and the code of DreamerV2, but the added term to maximize diversity among explorers in order to achieve better exploration / zero-shot generalization is interesting.

---

> ### Author Response · Authors · 2022-08-02
> **Thank you for your review! [1/2]**
>
> We appreciate your thorough review, providing us with plenty of opportunities for clarification and improvement. We believe we have addressed all of these concerns, so please let us know if anything else is required for your score to be improved.
>
> ### W1: Lack of experimental evidence.
>
> First, we respectfully disagree about our experiments not being on complicated environments. Concretely:
> * MiniGrid is a non-trivial exploration environment, used by state of the art model free algorithms explicitly testing for exploration given the sparse reward and partial observability. See for instance: RIDE (ICLR 2020), AMIGo (ICLR 2021), AGAC (ICLR 2021), NovelD (NeurIPS 2021) just to name a few. Our reward-free agents can solve two of the tasks in this benchmark zero-shot, only deploying a handful of exploration policies.
> * From Atari we include Montezuma’s Revenge which is widely considered to be one of the most challenging exploration environments. Our agents can get >0 performance zero-shot with reward-free exploration. We now also have additional Atari experiments, each presenting their own unique exploration challenges.
> * For DMC we use Walker which is one of three environments considered by the URLB benchmark (NeurIPS 2021). Note that in URLB the “unsupervised” methods do not perform especially well here, so this is very much still the frontier for reward-free deep RL research. Further, we consider initializing our agents with *three different datasets*: random, medium and expert. This is essentially three different experiments all using the walker benchmark. See Figure 7 in the Appendix for the full results. This alone is a large-scale experiment, using one of the environments widely studied for current SoTA reward-free RL algorithms.
>
> Second, since we appreciate that you may need further convincing, we have made our best effort to run new experiments *based on your specific recommendations*, which are viewable in the general response and updated manuscript. Concretely, we have been able to run 10 seeds for each method in Crafter. We tested each agent with 20 deployments with a deployment size of 50k, using population size of 10 for PP2E and CASCADE. This sums to 1M steps, for fair comparison to the original Crafter paper. We use the evaluation protocol from the paper (geomean from all training data) to produce the “Crafter Score” as follows:
>
> |         Method         | Random | P2E | PP2E | CASCADE |
> |:-------------------:|:--------:|:--------:|:------:|:------------:|
> | IQM          |  1.54 |    2.03    |   2.03 |   2.07  |
> | (sem)          |  (0.01)  |    (0.03)    |  (0.02)   |  (0.02)  |
>
> As can be seen, we do see gains here for CASCADE. Interestingly, PP2E does not outperform P2E, indicating that random initialization provides *insufficient* diversity. Meanwhile, CASCADE does see a small improvement, which we believe could be extended in future work. In particular, focussing on improved *behavioral representations*, which is an active area of work in the quality diversity, multi-agent and deep RL communities.
>
> Further, we have also added results on two of the Atari games mentioned: Hero and Frostbite; please see the updated paper for details about these (principally Figure 4).
>
> ### W2: Limited comparison to prior work
>
> We respectfully disagree that our current experiments are “classical toy environments”. There are many recent purely empirical papers in RL using these environments to demonstrate SoTA agents (e.g. RIDE, AGAC, NovelD). Instead, we use these same environments in a far more complex setting of reward free deployments to learn a general world model. To reiterate:
> - MiniGrid is widely used by SoTA exploration agents due to partial observability and sparse reward. Further it is procedurally generated.
> - Montezuma’s Revenge has been a huge focus for exploration works for the past 5-10 years. Very few works consider this in the reward-free setting.
> - DMC Walker is from a brand new benchmark (URLB) which only came out last year. It seems very unfair to call this a “classical toy environment” when it is less than 12 months old. Further, we use three different offline initializations, which was maybe missed by the reviewers.
>
> The criticism that the setting is not justified by real world settings seems unreasonable. Almost all deep RL papers at NeurIPS could be rejected on this basis. In fact, our setting is very much justified by real world settings. Think of the multiple robot arms working simultaneously to collect data. In this case B is small but not 1. Further, the majority of offline RL works simply use D4RL, a proprioceptive (and lower dimensional) benchmark which is far more toy than using a latent world model *from pixels* in three different experimental domains; indeed, the original deployment-efficiency paper uses this domain [64].

---

> > ### Author Response · Authors · 2022-08-02
> > **[2/2]**
> >
> > ### W3: Formalism unclear
> >
> > We hope this is now largely cleared up in the general response, but to answer your individual concerns, when moving to the trajectory perspective, we want to maximize over policies as these induce trajectories; note that Plan2Explore still takes an argmax, but instead over actions as their formulation focuses on the one-step setting.
> >
> > Regarding the intuition behind the mutual information between a distribution of states and an imagined MDP, when decomposing the objective into a difference in entropies, (i.e., Eq. 1 in our paper), we see that the first term refers to general diversity in the trajectories, whilst the second term refers to diversity in trajectories that is irreducible (since we condition on the MDP, e.g., as a result of inherent transition function stochasticity). Taking their difference gives the epistemic uncertainty over the trajectories themselves, hence why we choose to maximize this term. Then, when factoring in the population, as we do in Eq. 4, we need to make sure we don’t ‘double count’ the epistemic uncertainty due to those trajectories already being preferred by other individual policies.
> >
> > ### W4. Theory is motivation not guarantee
> >
> > “Theory is motivation not guarantee”... we agree with this statement! This is a deep RL paper, our theoretical statements should be seen as a motivating guide to justify the design choices we have made. We do not consider this work to be a theoretical paper nor we believe our main results are theoretical in nature; we have now changed the wording in the paper to reflect this.
> >
> > ### W5. Writing more polished
> > We have now made these changes in the paper, thank you for finding them.
> >
> > Moving to the questions....
> >
> > ### Q1: “While true”?
> >
> > In theory we could run this algorithm forever, it is intended to be open-ended. The key idea is that at each deployment we collect a large batch of data. This is very different to existing paradigms where the model often retrains every timestep with a single episode of new experience being collected at a time. That being said, we agree it is confusing in this specific context so we made a change in the manuscript. We hope it is clearer now.
> >
> > ### Q2: L.97 clarification
> >
> > There is no difference from Plan2Explore here in principle, it is just 1) the specific objectives used and 2) the number of steps collected with the subsequent policy. We collect thousands of steps vs. P2E which updates every timestep.
> >
> > ### Q3: Fig3
> >
> > The only answer we can give is that the train-time data collection may not be a perfect proxy for the generality of the world model. We tried to give a few different metrics to show the performance of different approaches for this very reason (see our reply to reviewer zCLK for additional details). The fact that CASCADE is pareto optimal here is reassuring that it is the strongest method.
> >
> > ### Q4: Fig3
> >
> > Following the previous question, it is simply the case these two metrics do not perfectly capture the breadth of the distribution of data collected at train time. It could be the case that the state coverage is the same but CASCADE is more uniform over the covered space so models it better.
> >
> > ### Q5: Fig5
> >
> > Interesting observation, we do not know any reason why this would be the case. The models are all trained on the same initial data.
> >
> > ### Q6: Addressing weaknesses
> >
> > Absolutely! Thank you for the opportunity :)

---

> > > ### Comment · Reviewer_Yjhj · 2022-08-07
> > > **Response to rebuttal**
> > >
> > > Thanks for your response. My concerns about toy environments were addressed, and the additional experiments on Atari and Crafter improve the overall results. I updated my score to 5.
> > >
> > > By the way, did you try Freeway, which is often considered as a significant exploration problem with sparse rewards?

---

> > > > ### Author Response · Authors · 2022-08-07
> > > > **Thank you for your response :)**
> > > >
> > > > Thank you for taking the time to read our response and for acknowledging that we addressed some of your concerns.
> > > >
> > > > Regarding Freeway, given compute constraints we only had one seed when the rebuttal period ended (vs. five for others) so did not manage to include it. We will now resume these experiments and maybe get a result before the discussion period ends (August 9th 1pm PT). Regardless of the outcome we will include the results in the CRC.
> > > >
> > > > We would appreciate it if you could help us understand what, if anything, you feel is holding you back from full support of the paper (i.e. a score of 6/7) given the concerns we've addressed.
> > > >
> > > > Thank you!

---

> > > > ### Author Response · Authors · 2022-08-08
> > > > **Freeway results**
> > > >
> > > > We are excited to share that we were able to get five seeds for Freeway faster than expected. The performance is very strong, as you can in the paper and pasted below:
> > > >
> > > > |         Method         |    Random |    P2E   |    PP2E |   CASCADE   |
> > > > |:--------------|:-------------:|:-------------|:-----------|:------------|
> > > > | IQM          |  1.06 |   6.36     |  17.36 |   29.22  |
> > > > | (95% CI)   |  ( 0.47, 1.84)  |  (3.66, 9.18)  |  (14.01, 20.67)  |  (28.88, 29.54)  |
> > > >
> > > > Briefly, we match human performance *zero-shot*, and perform nearly as well as methods that can solve this task (such as Ape-X, which gets 34.0).
> > > >
> > > > We want to extend our deepest thanks for all of the suggestions thus far: your feedback has made our work significantly stronger and we are now confident it would be a great contribution for NeurIPS. We hope that you feel your are in a position to consider supporting our paper for acceptance with a score of 7+, especially in light of the strong empirical results in the extended experiments you called for (thanks again!).
> > > >
> > > > Thank you!

---

### Official Review · Reviewer_zB57 · 2022-07-10

**Rating:** 6
**Confidence:** 5
**Soundness:** 4 excellent
**Presentation:** 3 good
**Contribution:** 2 fair

**Summary:**

This paper discusses a common problem in DRL, w.r.t, low efficient exploration in sparse reward tasks, and poor generalization of the trained agent. The authors propose CASCADE that learns to build a world model following a self-supervised exploration strategy. The naive motivation of CASCADE is to theoretically improve the learning objective of Plan2Explore to be more generalized. The results of the empirical evaluation show the effectiveness of the proposed method.

**Questions:**

**Concerns and Questions**

1. The pseudo-code leads to misunderstanding. There is only one exploration policy in algorithm 1, which does not match the main contribution of this paper. The input of algorithm 1 should be multiple policies, which can be more explicit and reinforce the proposed method's main difference from the others.

2. As mentioned in Line.123, the embedding of the representation is the final state. I wonder what the advantages of doing this are. It seems that there are multiple MDPs existing. Why not embed the representation in terms of similarity or uncertainty.

3. There is a typo in Line 136. The proof is in Appendix C.1.1.

4. There lacks an explanation of why it is necessary to extend eq.2 to the population-based version (eq.3).

5. It would be great to include the evaluation results of CASCADE in more complicated environments (as mentioned above). It does not matter even if the trained policy's performance is not very good.


**Limitations:**

**Limitations and potential negative societal impact**

N/A

**Reference**

[1] Song, S., Yu, F., Zeng, A., Chang, A. X., Savva, M., and Funkhouser, T. (2017). Semantic scene completion from a single depth image. In Proceedings of the IEEE Conference on Computer Vision and Pattern Recognition, pages 1746–1754.

##upload after the rebuttal##

Considering that the authors have conducted a lot of work during the rebuttal, the added experiments reinforce the empirical evaluation parts of the paper in a way. I raise my score by 1.

**Strengths And Weaknesses:**

**Strengths:**

1. This paper is well structured and easy to read. The motivation is easy to understand, and I agree with it. It seems that the authors are not trying to over-sell the contributions, which is good.

2. The theoretical proof is clear, and there is no mistakes in term of notations.

3. The representation of the experiment results is clear, and it is clear to find how CASCADE works.

4. This paper includes a strong background introduction and related research.

**Weakness**

1. I encourage the authors to improve the part of the abstract. I cannot tell precisely the novelty and necessity of CASCADE in the first reading.

2. The experiments are not strong enough to demonstrate the advantages of the proposed method. I was expected to see CASCADE to work in more complicated environments, e.g., the SUNCG dataset (refer to [1]).

---

> ### Author Response · Authors · 2022-08-02
> **Thank you for your review! [1/2]**
>
> Thank you for your positive review, we appreciate that you found the motivation easy to understand and the paper to be clear. It seems your concerns are twofold: 1) the abstract quality 2) more “complicated” environments. We will try to address these below.
>
> ### 1. The abstract.
>
> We completely agree; we should have done a better job specifying the novelty and necessity of CASCADE. We have subsequently changed the abstract based on the reviewer’s feedback and we hope that it is now much clearer!
>
> ### 2. a) The experiments are not strong enough, b) need more complicated environments.
>
> a) We are always seeking to improve our work and we have made our best effort to run some new experiments to strengthen the paper. Based on feedback from reviewers we are pleased to let you know that we have been able to run 10 seeds for each method in Crafter. We tested each agent with 20 deployments with a deployment size of 50k, using population size of 10 for PP2E and CASCADE. This sums to 1M steps, for fair comparison to the original Crafter paper. We use the evaluation protocol from the paper (geometric mean from all training data) to produce the “Crafter Score” as follows:
>
> |         Method         | Random | P2E | PP2E | CASCADE |
> |:-------------------:|:--------:|:--------:|:------:|:------------:|
> | IQM          |  1.54 |    2.03    |   2.03 |   2.07  |
> | (sem)          |  (0.01)  |    (0.03)    |  (0.02)   |  (0.02)  |
>
> As can be seen, we do see gains here for CASCADE. Interestingly, PP2E does not outperform P2E, indicating that random initialization provides *insufficient* diversity. Meanwhile, CASCADE does see a small improvement, which we believe could be extended in future work. In particular, focussing on improved *behavioral representations*, which is an active area of work in the quality diversity, multi-agent and deep RL communities.
>
> Further, we have also added results on two of the Atari games mentioned: Hero and Frostbite, as well as including more deployments for DMC. In all cases CASCADE demonstrates an improvement over the baselines. Please see the updated paper and general response for more detail.
>
> b) We have to disagree about our experiments not being complicated environments. Concretely:
> * MiniGrid is a non-trivial exploration environment, used by state of the art model free algorithms explicitly testing for exploration given the sparse reward and partial observability. See for instance: RIDE (ICLR 2020), AMIGo (ICLR 2021), AGAC (ICLR 2021), NovelD (NeurIPS 2021) just to name a few. Our unsupervised agents can solve two of the most difficult tasks in this benchmark zero-shot.
> * From Atari we include Montezuma’s Revenge which is widely considered to be one of the most challenging exploration environments. Our agents can get >0 performance zero-shot with reward-free exploration. We now also have additional Atari experiments, each presenting their own unique exploration challenges.
> * For DMC we use Walker which is one of three environments considered by the URLB benchmark (NeurIPS 2021). Note that in the URLB the unsupervised methods do not perform especially well here, so this is very much still the frontier for reward-free deep RL research. Further, we consider initializing our agents with *three different datasets*: random, medium and expert. This is essentially three different experiments all using the walker benchmark. See Figure 7 in the Appendix for the full results. This alone is a large-scale experiment, using one of the environments widely studied for current SoTA reward-free RL algorithms.
>
> We argue that each of these environments is non-trivial and highly relevant for the current frontier of reward-free RL. The key point though is that we have a single method that works on all of them with *no implementation modifications*, demonstrating our method’s generality.

---

> > ### Author Response · Authors · 2022-08-02
> > **[2/2]**
> >
> > Moving to the Concerns/Questions:
> >
> > ### Q1. Pseudocode exploration policy.
> > This is to improve the generality of the formulation, and CASCADE still naturally fits into this. We can consider CASCADE as comprising a *single policy* that can switch between B behaviors during deployment without retraining. We then collect A * B episodes at deployment time, since we collect A episodes with each of the B behaviors. This could be conducted either in parallel or sequentially. We have made a comment on this in the paper as we agree it was not clear before.
> >
> > ### Q2. Embedding choice.
> > This is a great question, and it is very much an open problem in the literature. It boils down to the following: “what is the right behavioral representation with which to measure diversity?”, which we have seen discussed in a huge range of fields in RL. In our case, we take the final recurrent state for two main reasons. First, inspired by work in neural machine translation [cite Seq2seq], which use the final encoder RNN state to decode complex translations, we note that the final state should contain information that summarizes the trajectory. Therefore if two final states are similar, it is reasonable to assume that their imagined trajectories are also similar; we find this to be the case empirically. Second, it is significantly more tractable to compare only final states between agents; indeed we experimented trying to utilize non-parametric distance estimation of imagined trajectory latents, and observed that these did not perform as well, and took significantly longer to run.
> >
> > However, we fully accept that if we want to solve something more complex, such as Crafter, this may need a more sophisticated embedding which explicitly models longer term interactions. We leave this for future work.
> >
> > ### Q3. Typo
> > Fixed, thank you!
> >
> > ### Q4. Jump from Eq 2 -> 3.
> >
> > As identified, in our method we produce $B$ exploration policies. This makes it paramount we extend the single policy mutual information maximization formulation that produces a single policy (equation 2) to the multi policy setting (Equation 3). As we show (Lemma 1) it is not sufficient to produce a single policy and play it $B$ times. We hope this is clearer now!
> >
> > ### Q5. More complex experiments.
> >
> > See the response above, we have some exciting new results! We are glad that the reviewer agrees the method doesn’t have to work perfectly in every environment, but in many cases it does provide an improvement.
> >
> > To conclude, we have now improved the abstract by making our contribution clearer, and we believe our experiments were already thorough and general but have improved them anyway. Given this response, we kindly hope the reviewer provides additional support for our paper by increasing to an accept.

---

> > ### Comment · Reviewer_zB57 · 2022-08-07
> > **Thank you for reponses.**
> >
> > Thank you for efforts in rebuttal. I think the authors have addressed my main concerns in the response. I agree that this paper proposes an attractive solution for one critical problem in the community, and the empirical evaluation proves it works. However, although you have upgraded some parts of the paper, I believe that more proofreading can better tell your story. I tend to keep my score.

---

> > > ### Author Response · Authors · 2022-08-07
> > > **Please reconsider given concerns have been addressed :)**
> > >
> > > We appreciate you taking the time to read our response and for commenting that we have addressed all of your concerns. We have to say, it is very confusing to see such positive comments without an increased score, especially given we are currently only at a 5.
> > >
> > > In particular, you note we have an “attractive solution for one critical problem”, with the only concern remaining being “proofreading”, yet in the initial review the first strength was “This paper is well structured and easy to read” and we will have an additional page in the camera ready and can easily spend time making the paper more coherent if given the chance.
> > >
> > > Please can you either increase or provide us with a genuine reason why the paper does not warrant accepting?

---

> > > > ### Comment · Reviewer_zB57 · 2022-08-08
> > > > **Reply to your quandary**
> > > >
> > > > I am sorry for the confusion. Maybe I should give you a clearer explanation for my rating. The point "5" is kind of "accept" rather than "reject".
> > > >
> > > > As I mentioned, the core idea of your work is attractive. It does focus on one critical problem of the RL community.
> > > >
> > > > However, a good idea is not equal to a good story. There may exist some misunderstanding of my comments. That is, the "well-structured" and "easy-to-read" actually are for the technical parts (sec.3) of the paper. I believe this paper will benefit a lot from re-writing the abstract/introduction parts.
> > > >
> > > > Meanwhile, I note that you have greatly upgraded the analysis parts (sec.4) and added some experiments in the rebuttal. However, the first added experiments were not enough for me to raise the score, and I didn't ask for more complex environments due to the limited time. I want to see how CASCADE works in a more realistic environment or image-input tasks, and that's one of the most direct ways where the "world model" shows its true value and meaning to the RL community.
> > > >
> > > > Considering you have conducted a lot of work in the rebuttal, I can raise my score by 1. BUT remember to upgrade the writing and tell a better story. If possible, add some more realistic experiments to show the advantages. Good luck.

---

> > > > > ### Author Response · Authors · 2022-08-08
> > > > > **Thank you for your response :)**
> > > > >
> > > > > Thank you for this! We definitely agree, now that we have much stronger empirical results the intro is likely the place needing the most work if our paper is going to achieve the maximum possible impact. We will for sure take this on board and focus on improving it for the camera ready, which should be possible given we get an additional page.
> > > > >
> > > > > Regarding our experiments, we want to flag that you mentioned *"I want to see how CASCADE works in a more realistic environment or image-input tasks"*, but all of the environments in this paper are image input aside from Minigrid, which uses a similar type of observation (just not rgb). Crafter/DMC/Atari are all from pixels, and for this reason our method is based on DreamerV2. These are reasonably large-scale experiments!
> > > > >
> > > > > With this in mind, would you feel your are in a position to give our paper an "accept", rather than "weak accept"?
> > > > >
> > > > > Thank you!

---

### Official Review · Reviewer_zCLK · 2022-07-11

**Rating:** 6
**Confidence:** 4
**Soundness:** 3 good
**Presentation:** 3 good
**Contribution:** 3 good

**Summary:**

This work introduces a new problem setting, Reward Free Deployment Efficiency focusing on two incentives: 1. Task agnostic exploration facilitates generalization. 2. Exploration policies that can collect large quantities of data without centralized retraining facilitate scalability.

In addition, this work introduces the Coordinated Active Sample Collection via Diverse Explorers (CASCADE), which can gather a diverse set of data and is inspired by Bayesian Active Learning.

In this work, B exploratory agents are trained in parallel. At each deployment, a loss containing two parts is maximized; the first part is a diversity term between the agents' behaviors. And the second part is the so-called information gain.

The authors have used the DreamerV2 agent as the base agent for their work and their baselines. Furthermore, they used the well-known Plan2Explore (P2E) and a modified version (Population Plan2Explore, PP2E) as their baseline. Finally, they have produced experiments to show how well their proposed method can explore the state space and also perform with a zero-shot manner to the task-specific problem.

**Questions:**

1. Why is rewarding-episodes being considered in a reward-free setting? (This is a bit confusing)
2. Would it be possible to have the final performance of the P2E (not in the deployment efficiency setting) reported as a dashed line? This is informative in case one wants to see how difficult the proposed problem setting is.
3. Could you elaborate more on this?

>to facilitate scalability, exploration policies should collect large quantities of data without costly centralized retraining.

 I can easily imagine real-world scenarios where access to about 20 different agents is not possible.


**Limitations:**

The authors have addressed the limitation of their work in section 4.3.

**Strengths And Weaknesses:**

Originality:
This work introduces a new setup that is called reward-free deployment efficiency. Furthermore, inspired by the P2E agent, the authors propose a new method called CASCADE that can perform decently compared to the previous baseline in this setup.

Quality:
The authors have done an excellent job explaining their setup, proving their claims, and discussing their experiments.

Clarity:
The paper is well-written which prevents a familiar reader from doing additional passes through different paragraphs.

Significance:
Despite the fact that the proposed method, CASCADE, is well-motivated, I do think that there are few arguments about why the reward-free deployment efficiency matters. I can assume real-world scenarios where access to about 20 different agents is not possible.

---

> ### Author Response · Authors · 2022-08-02
> **Thank you for your review!**
>
> Thank you for your positive review! We are pleased to see you found the work to be well-motivated and high quality. Focusing on the Questions and Weaknesses, it seems the largest area of concern can be addressed with better explanation on our side. We will attempt to explain here and we hope that if this satisfies the concerns you will consider raising your score.
>
> ### Why is rewarding episodes being considered in a reward-free setting?
>
> This is a great question, we agree it is confusing. Essentially we are trying to measure the *overall effectiveness* of the exploration policy. We decided that a fair way to measure would be “reward” + “coverage” because:
> * *Rewarding episodes* may capture the discovery of more complex behavior, but does not indicate the *breadth* of exploration. For instance, consider an agent that adopts a complex exploration policy that consistently reaches the furthest room, but does not explore any rooms in between. This would facilitate the learning of a *narrow* set of complex behaviors, but will struggle if the goal gets moved at test time to a room that’s closer to the initial position.
> * *Coverage* will reward filling the space but may not capture depth of exploration or more complex behaviors to get the final few percentage points. For instance, an agent may learn to fully cover an entire room, but not learn complex behaviors such as opening doors and entering new rooms.
>
> We therefore believe that considering these two metrics together provides a reasonable representation of the exploration effectiveness. We then use zero-shot transfer performance as a further validation of the quality of the model. Finally, we note that recent exploration works also use similar metrics when evaluating [1, 2].
>
> ### Final performance of P2E.
>
> We will try to add this; it may not be ready for the rebuttal but will be in the CRC. For sure we will see better performance for P2E in the DMC tasks.
>
> ### Elaborating on collecting without retraining.
>
> This is largely addressed in the general response, but essentially imagine having a finite amount of time to collect data with a policy; we would want that policy to do diverse things to facilitate learning unknown downstream tasks. Essentially our method will switch between pre-trained diverse behaviors, rather than constantly deploy the same behavior. PP2E of course does the same thing, but without explicitly enforcing diversity in the behaviors.
>
> Thank you again - please also check out the additional experiments (from the other reviewers) that are shared in the individual responses but more clearly in the general response and updated manuscript. Please let us know if there is anything else we need to address for you to raise your score.
>
> [1] Flet-Berliac et al. Adversarially Guided Actor-Critic. ICLR 2021
>
> [2] Zha et al. Rank the Episodes: A Simple Approach for Exploration in Procedurally-Generated Environments. ICLR 2021

---

> > ### Author Response · Authors · 2022-08-08
> > **Follow up**
> >
> > Hi Reviewer zCLK,
> >
> > We understand reviewer load is high and we thank you again for your time!
> >
> > We just wanted to flag that we have made significant improvements to our paper, with new experiments and additional clarifications (based partly on your specific recommendations). Other reviewers have now raised their scores, and we were hoping you might consider doing the same, given you were originally "borderline accept" and our paper is now much stronger.
> >
> > Thank you!

---

> > > ### Comment · Reviewer_zCLK · 2022-08-09
> > > **Thanks for the Response**
> > >
> > > I want to thank the authors for their great use of the rebuttal. My main concerns are justified. I am still a bit skeptical about the rewarding episodes; however, it might be just some personal preference.
> > > I have raised my rating by 1 point.

---

> > > > ### Author Response · Authors · 2022-08-09
> > > > **Thank you for your response!**
> > > >
> > > > Hi Reviewer zCLK,
> > > >
> > > > Thank you for coming back, and for increasing your score to a "weak accept". It seems your only remaining concern is regarding the use of rewarding episodes as a metric for evaluating exploration. We want to reiterate that it is just being used as a proxy for depth of exploration, which we combine with state coverage to show breadth of exploration. We will expand on this in the additional page for our CRC, alongside re-wording sections of the intro and some Crafter analysis.
> > > >
> > > > However, given the confusion (which we do think is reasonable) we actually removed the rewarding episodes plot from the main body for Montezuma's Revenge. Instead, we show the zero-shot performance for our Atari results. If you have a spare moment then please check out the revised paper to see how it looks.
> > > >
> > > > In light of this, we hope that you feel your are in a position to consider supporting our paper for acceptance with a score of 7+.
> > > >
> > > > Thank you!

---

### Author Response · Authors · 2022-08-02
**General Response + Paper Update [1/2]**

We thank the reviewers for taking the time to provide us with thorough feedback. Overall we believe the reviews to be constructive and we have made every effort to address all concerns, including running a large variety of additional experiments and providing clarifications. **We have updated our paper to include these changes and believe it led to a much stronger version of the paper**. Given that our initial review scores were generally borderline, we hope to see the reviewers’ support for our paper to increase (to accept) or to receive additional feedback on how to further improve the paper.

Please see below some highlights of common themes in the responses, changes to the paper and new experiments.

### How come we used B policies? Is this practical?

We assume this refers to the population of B agents. One area of confusion (which we hope we have clarified in the updated paper) is regarding the fact that we have an exploration policy consisting of B behaviors. One way to view this is that it is *one policy*, but can deploy different behaviors at different times. Concretely, consider it as a single policy with multiple behaviors “pre-loaded”. When might this be practical? Here are a few examples:
* If you have a fleet of robots, for example a room full of robot arms, each arm could index a different behavior from the exploration policy. This is a common use-case in robotics, where data collection on a single robot is often impractical [46].
* If you have a single robot but access for a few hours, each time there is a reset the behavior could switch.
* If you are hoping to collect data with parallel compute, it may be possible to collect A episodes in parallel with A >> B. We would then collect A/B episodes with each of the B behaviors. It does not make sense to do all of this collection with the *same* behavior policy as most of the experience will be a duplicate. For instance, with a single GPU in a parallel simulator like Brax, A = 2000, so if we have a population of B=20 we are still collecting 100 episodes with each behavior. This is much better than having 2000 episodes with the *same* behavior which leads to a homogenous dataset that will not aid the generalization of our world model. Our work is the first step towards being able to appropriately leverage this type of simulator for learning world models.

In all of these cases we do not want to wait minutes to hours to retrain new behaviors. So having access to a diverse set of pre-trained behaviors makes it possible to collect a rich dataset in a single “deployment”.

### How are ImagDiv and InfoGain related and how are they distinct?

We see that several reviewers have expressed a lack of clarity regarding these two terms, and fully accept our failure to communicate this effectively in the original manuscript. In short, InfoGain is the epistemic one-step dynamics uncertainty maximization term from Plan2Explore, and ImagDiv is a new novelty seeking term whose formulation is partly *inspired* by that of Plan2Explore, but is distinct by maximizing overall *trajectory* diversity, not maximizing one-step entropies.

To better understand this, consider the original InfoGain term in Plan2Explore (i.e., Eq 5 in their work). Concretely, this term relies on maximizing mutual information between the next latent state and parameters *conditioned on the current state and action*. In comparison, our ImagDiv formulation *does not* condition on the current state and action, and instead looks at the entire trajectory, which it does through the embedding function $\Phi$.

Why is it important that our information theoretic formulation is not per-state? First, defining a per-state information gain objective in the way Plan2Explore does is suboptimal for deep exploration. This is because such objectives encourage local exploration, which reduces the uncertainty at a given state, but may not result in the best reduction in uncertainty in discovering the structure of the whole MDP, particularly in the batch deployment setting we study in this work.

To further understand this, the generalization of the InfoGain objective in Plan2Explore would reduce to stitching together per-state entropy maximizing policies that can maximize the per state epistemic coverage, but may fail at maximizing the *global structure* coverage of the MDP. For example, an ensemble of policies as defined in a tree MDP could therefore avoid each other on a per state basis, but may not maximize the probability of having all policies end up at a different leaf, crucial for ensuring policies that achieve diversity in the environment when taken together. To this latter point, we exploit the submodularity of the population formulation to ensure this ‘deep’ diversity in the trajectories.

---

> ### Author Response · Authors · 2022-08-02
> **[2/2] Including new experiments!**
>
> Having said this, we empirically find that there is still a benefit to also incorporating the InfoGain objective, and as such introduce a trade-off parameter that controls how much we favor the more ‘local’ uncertainty reduction exploration that InfoGain encourages, and the more  ‘global’ uncertainty reduction exploration that ImagDiv encourages.
>
> We have tried to make this clearer in the updated manuscript, but with the benefit of another page for the camera-ready, we will definitely include these finer details.
>
> ### Aren’t these toy experiments?
>
> In short, no! The experiments we include are in all environments still commonly used by SoTA deep RL papers. In particular:
> * MiniGrid is partially observable, procedurally generated and has sparse rewards. It is regularly used for SoTA exploration algorithms, as can be seen [here](https://github.com/Farama-Foundation/gym-minigrid).
> * Montezuma’s Revenge has been used for deep RL exploration methods for the past 5-6 years. It remains a challenge for many methods, even with rewards. By operating in the reward-free domain, this becomes even more challenging (as the sparse rewards still provide indicators of progress).
> * Walker was recently proposed as a benchmark for unsupervised RL in URLB. This came out at NeurIPS 2021, less than a year ago. We also use three different settings with differing offline dataset initializations, providing an interesting use-case showing our method can be subsequently deployed with any initial data and improve generality. This is clearly not a toy benchmark!
>
> Further, we consider a *more challenging* setting for all of these benchmarks, using just a handful of deployments (rather than the fully online setting in most prior work). Further still, we use the current SoTA world model, DreamerV2. Finally, these experiments are distinct, and crucially without changing *any implementation details*, our method works well in all settings.
>
> ### New experiments
>
> Even though we feel our existing experiments are thorough and sufficient, we think some ideas from the reviewers are worth exploring to provide additional evidence for the efficacy of CASCADE. We therefore present new results, which are all included in our revised manuscript. In particular, we focus on the Crafter environment, highlighted by both reviewers zB57 and Yjhj. As alluded to, Crafter is a highly challenging environment for reward-free RL algorithms, and P2E is the state of the art agent in the online setting here.
>
> We have been able to run 10 seeds for each method for 20 deployments using a deployment size of 50k and a population size of 10. This represents 1M steps, as included in the original Crafter paper. We use the evaluation protocol from the paper (geomean from all training data) to produce the “Crafter Score” as follows:
>
> |         Method         | Random | P2E | PP2E | CASCADE |
> |:-------------------:|:--------:|:--------:|:------:|:------------:|
> | IQM          |  1.54 |    2.03    |   2.03 |   2.07  |
> | (sem)          |  (0.01)  |    (0.03)    |  (0.02)   |  (0.02)  |
>
> Indeed, we are pleased to report that we do see gains here for CASCADE. Interestingly, we do not see any gains from PP2E, indicating that random initialization provides insufficient diversity. Meanwhile, CASCADE does see a small improvement, which we believe could be extended in future work. In particular, focussing on improved *behavioral representations*, which is an active area of work in the QD, multi-agent and deep RL communities.
>
> #### **DMC, more deployments**
>
> We wanted to see how our methods compare on DMC with more data. In the original paper we showed the performance after 1 and 2 deployments. We have now expanded these results to *15 deployments*. Once again we see consistent gains for CASCADE vs. the baselines. See the paper for the new expanded results.
>
> #### **New Atari Experiments**
>
> Based on the feedback from Reviewer Yjhj we have been able to run two additional Atari environments. We note that these results are only for five seeds and not extensively tuned, which should be expected given the fast turnaround. Nonetheless, we once again see that CASCADE outperforms other methods in these domains. We have revamped our paper to emphasize the zero-shot performance, since other reviewers found the training metrics misleading. These will be in the Appendix in the camera ready, for completeness. We now have a new Figure  4 that includes *three Atari games* and Crafter, vs. previously just one Atari game. This represents a significantly more exhaustive set of experiments which alone we believe is sufficient for improved review scores. As we discuss in the updated manuscript, we see that CASCADE is statistically significantly better than the baselines in these new environments, and furthermore consistently displays strong performance across all environments.

---

### Author Response · Authors · 2022-08-08
**More new results!**

Hi all,

Before the discussion period comes to a close, we would like to ask for a few more moments of your time. Based on the request of Reviewer Yjhj, we ran further experiments on a fourth Atari game, Freeway, and the results are now in the paper. As you can see below, CASCADE very strong here, matching human performance, and performing nearly as well as methods that can solve this task (such as Ape-X, which gets 34.0):

|         Method         |    Random |    P2E   |    PP2E |   CASCADE   |
|:--------------|:-------------:|:-------------|:-----------|:------------|
| IQM          |  1.06 |   6.36     |  17.36 |   29.22  |
| (95% CI)   |  ( 0.47, 1.84)  |  (3.66, 9.18)  |  (14.01, 20.67)  |  (28.88, 29.54)  |

In the camera ready version we will have an additional page, so can further elaborate on these results and provide some additional analysis for Crafter. We can also use this space for some additional explanation of our method and to add more citations.

We are pleased to see there have been some upgrades from reviewers, with all now in favor of acceptance. It would be great if the reviewers could provide further support given the strength of our new results (3 new Atari games + Crafter + more DMC deployments) and additional clarifications made in the paper (in red).

Thank you and have a great day!

---

### Meta-Review · Area_Chair_EZn7 · 2022-08-30

**Recommendation:** Accept
**Confidence:** Less certain

**Metareview:**

This paper proposes a method to learn world models without rewards, using a collection of agents that explore an environment. The key idea is to maximize diversity between the trajectories collected by the agents to obtain a good world model, with an emphasis on being as efficient as possible. The authors present some theoretical justification for using a population of agents and their empirical results on several datasets provide a good demonstration of the method. The reviewers all agree this is an interesting and important setting and the author response significantly improves the paper on aspects of clarity and empirical results, based on the reviewer concerns. Overall, I believe this work provides interesting ideas and will encourage more work in this direction in the future. I encourage the authors to revise their paper taking the reviewer suggestions into account and add in the new experiments to make it stronger.

**Award:**

No

---

### Decision · Program_Chairs · 2022-09-14

Accept